



# Disentangling the impact of air-sea interaction and boundary layer cloud formation on stable water isotope signals in the warm sector of a Southern Ocean cyclone

Iris Thurnherr[1,2] and Franziska Aemisegger[1]

[1]Institute for Atmospheric and Climate Science, ETH Zürich, Zurich, Switzerland
[2]Geophysical Institute, University of Bergen, and Bjerknes Centre for Climate Research, Bergen, Norway

**Correspondence:** Iris Thurnherr (iris.thurnherr@uib.no)

**Abstract.** Stable water isotopes in marine boundary layer water vapour are strongly influenced by the strength of air-sea fluxes. Air-sea fluxes in the extratropics are modulated by the large-scale atmospheric flow, for instance by the advection of warm and moist air masses in the warm sector of extratropical cyclones. A distinct isotopic composition of the water vapour in the latter environment has been observed over the Southern Ocean during the 2016/17 Antarctic Circumnavigation Expedition (ACE). Most prominently, the secondary isotope variable deuterium excess ($d = \delta^2 H - 8 \cdot \delta^{18} O$) shows negative values in the cyclones' warm sector. In this study, three mechanisms are proposed and evaluated to explain these observed negative $d$ values. We present three single-process air parcel models, which simulate the evolution of $\delta^2 H$, $\delta^{18} O$, $d$ and specific humidity in an air parcel induced by decreasing ocean evaporation, dew deposition, and upstream cloud formation, respectively. Simulations with the isotope-enabled numerical weather prediction model COSMO$_{iso}$, which have previously been validated using observations from the ACE campaign, are used to (i) validate the air parcel models, (ii) quantify the relevance of the three processes for stable water isotopes in the warm sector of the investigated extratropical cyclone, and (iii) study the extent of non-linear interactions between the different processes. This analysis shows that we are able to simulate the evolution of $d$ during the air parcel's transport in a realistic way with the mechanistic approach of using single-process air parcel models. Most importantly, we find that decreasing ocean evaporation, and dew deposition lead to the strongest $d$ decrease in near-surface water vapour in the warm sector and that upstream cloud formation plays a minor role. By analysing COSMO$_{iso}$ backward trajectories we show that the persistent low $d$ observed in the warm sector of extratropical cyclones are not a result of material conservation of low $d$. Instead, the latter Eulerian feature is sustained by the continuous production of low $d$ values due to air-sea interactions in new air parcels entering the warm sector. These results improve our understanding of the relative importance of air-sea interaction and boundary layer cloud formation on the stable water isotope variability of near-surface marine boundary layer water vapour. To elucidate the role of hydrometeor-vapour interactions for the stable water isotope variability in the upper parts of the marine boundary layer, future studies should focus on high resolution vertical isotope profiles.





# 1 Introduction

Extratropical cyclones and their associated fronts are important weather features for moisture cycling over the Southern Ocean
(Wernli and Schwierz, 2006; Simmonds et al., 2012; Papritz et al., 2014). Horizontal temperature advection in the cold and
warm sector of extratropical cyclones leads to contrasting properties in the two sectors. The cold sector is dominated by equa-
torward transport of cold and dry air, which leads to intense large-scale ocean evaporation (Bond and Fleagle, 1988; Boutle
et al., 2010; Aemisegger and Papritz, 2018) and unstable conditions in the marine boundary layer (MBL, Beare, 2007; Sinclair
et al., 2010). In the warm sector, moist and warm air is transported polewards. The conditions in the warm sector are close
to saturation, which can lead to weak or even negative surface heat fluxes and generally a stable MBL (Beare, 2007; Sinclair
et al., 2010). These contrasting conditions in the cold and warm sector are also reflected in the MBL water vapour's isotopic
composition, which is a powerful tracer of moist atmospheric processes. Stable water isotopologues (SWIs) of water vapour are
altered during phase changes due to so-called isotopic fractionation and, thus, a distinct isotopic signal is imprinted in the water
vapour by the moist diabatic processes that occur along its transport pathway. Understanding the formation of SWI anomalies
in the cold and warm sector of extratropical cyclones, thus, gives important insight into the dominant processes affecting the
moisture budget of air parcels associated with extratropical cyclones.

The abundance of SWIs is given by the $\delta$ notation, which is defined by the relative deviation of the isotopic ratio of the sample
from an internationally accepted standard ratio representing the ocean water isotope composition: $\delta^2\mathrm{H}\,[\permil] = (\frac{^2R_{\mathrm{sample}}}{^2R_{\mathrm{VSMOW2}}\cdot 2} -$
$1)\cdot 1000$ and $\delta^{18}\mathrm{O}\,[\permil] = (\frac{^{18}R_{\mathrm{sample}}}{^{18}R_{\mathrm{VSMOW2}}} - 1)\cdot 1000$. $\mathrm{R}_{\mathrm{sample}}$ is the isotopic ratio of the water vapour defined as the ratio between
the minor (heavy) to the major (light) isotope. $^2R_{\mathrm{VSMOW2}}$=1.5576$\cdot 10^{-4}$ and $^{18}R_{\mathrm{VSMOW2}}$=2.0052$\cdot 10^{-3}$ define the isotopic
composition of the Vienna standard mean ocean water (VSMOW2). The strength of isotopic fractionation depends on environ-
mental conditions. Ambient temperature is the main factor affecting equilibrium fractionation that occurs due to the different,
temperature-dependent saturation vapour pressure of SWIs. Non-equilibrium fractionation, a second kind of isotopic fraction-
ation, occurs in non-equilibrium conditions, additionally to equilibrium fractionation, in the presence of humidity gradients
due to differences in the molecular diffusion velocity of the water isotopologues. For example, humid air that is oversaturated
with respect to the sea surface temperature (SST) can experience dew deposition on the ocean surface, which is accompa-
nied by non-equilibrium fractionation due to the humidity gradient towards the ocean surface. The secondary isotope variable
deuterium excess ($d$=$\delta^2$H-8$\cdot\delta^{18}$O) is a measure of non-equilibrium fractionation and therefore a tracer of diffusion processes
involved in ocean evaporation and dew deposition.

In the extratropics, the SWI abundance in MBL water vapour is shaped by various processes that are related to extratropical
cyclones. The strongest short-term SWI variations are observed during frontal passages. The passage of a cold front leads to a
decrease in $\delta$-values (Gedzelman and Lawrence, 1990), which is caused by below-cloud processes, such as rain evaporation,
equilibration of rain and water vapour, and by the horizontal advection of air masses with low $\delta$-values behind the cold front
(Pfahl et al., 2012; Aemisegger et al., 2015; Graf et al., 2019). Fewer studies are available on the isotopic variability in water
vapour during warm front passages. Gedzelman and Lawrence (1990) showed, that $\delta^{18}$O increases across the warm front due
to the advection of isotopically enriched air masses in the warm sector. These strong changes in SWIs across fronts reflect, on



the one hand, the precipitation-related processes along the fronts and, on the other hand, the contrasting properties and origin of water vapour in the cold and warm sector of extratropical cyclones, respectively, with typically high $\delta$ values in the warm sector and low $\delta$ values in the cold sector (Dütsch et al., 2016; Thurnherr et al., 2021).

Moisture transport over the Southern Ocean can occur over long distances. Precipitation in Antarctica can originate from sub-
tropical regions due to the suppression of ocean evaporation in the warm sector of extratropical cyclones (Terpstra et al., 2021). During this long-range transport, $d$ can potentially change due to non-equilibrium processes or changes in the ambient temperature (Dütsch et al., 2019). A detailed understanding of processes that potentially influence $d$ during transport in the warm sector is therefore needed. Low or negative $d$ has been observed in near-surface water vapour in the warm sector of Southern Ocean cyclones (Thurnherr et al., 2021), which is in strong contrast to high $d$ observed in the cold sector of extratropical cyclones
due to strong ocean evaporation (Gat et al., 2003; Uemura et al., 2008; Aemisegger and Sjolte, 2018). Vertical air-borne SWI profiles showed that negative $d$ can be seen also close to the boundary layer top and has been associated with the evaporation of cloud and rain droplets (Sodemann et al., 2017; Salmon et al., 2019). For near-surface water vapour, three processes have been identified based on a statistical analysis, which can lead to low $d$ in the warm sector of extratropical cyclones: dew deposition on the ocean surface, decreasing ocean evaporation and cloud formation as a result of moist adiabatic ascent (Thurnherr
et al., 2021). In this study, we investigate the role of these three processes in a quantitative way, for a selected Southern Ocean cyclone. A mechanistic approach is used to better understand the lifetime and characteristics of low $d$ in near-surface water vapour of the warm sector.

In situ measurements of SWIs are limited in time and space. Therefore, modelling approaches of various complexity have been often used to analyse the large-scale setting and to identify the underlying processes driving the measured SWI signals'
variability. Isotope-enabled numerical weather and climate models (e.g. Joussaume et al., 1984; Blossey et al., 2010; Werner et al., 2011; Pfahl et al., 2012; Eckstein et al., 2018) allow for detailed studies of the evolution of SWIs during transport (Dütsch et al., 2018; Dahinden et al., 2021). In addition to these complex numerical models, single-process models allow to investigate the factors controlling specific processes and their effect on the isotopic composition of water vapour. Different approaches have been used to model the isotopic composition of MBL water vapour of which many involve the Craig and Gordon (1965)
model (denoted CG65 hereafter). The CG65 model is a process model simulating the isotopic composition of water vapour that evaporates from a water reservoir. The comparison of measured SWIs in the MBL water vapour with the simulated isotopic composition of evaporated water vapour using the CG65 model has proven useful to estimate the contribution of ocean evaporation to the MBL water vapour (Benetti et al., 2014). Benetti et al. (2018) showed that the isotopic composition of water vapour in the subtropical MBL is a result of mixing of water vapour from ocean evaporation with dry air from the free troposphere
using a linear mixing model combined with the CG65 model. Adopting a Lagrangian point of view, Pfahl and Wernli (2009) followed air parcel trajectories experiencing ocean evaporation over the Mediterranean and modelled the isotopic composition of the water vapour using the CG65 model. Such an air parcel model (APM) allows to simulate the temporal evolution of SWIs in water vapour during the transport of an air parcel due to specific processes. In principle, APMs can also be designed for other processes such as dew deposition or cloud formation. In this study, we develop three single-process APMs to simulate
processes potentially influencing SWIs in the warm sector. We confront the results from these APMs with measurement and





a simulation with a SWI-equipped regional weather prediction model to understand the relative importance of ocean evaporation, dew deposition, and cloud formation for the isotopic composition of near-surface water vapour in the warm sector of an extratropical cyclone.

This study is structured in the following way: In Sect. 2, the used data sets are described. The single-process APMs are introduced in Sect. 3 and applied in a case study of a Southern Ocean cyclone in Sect. 4. Section 5 presents the conclusions of this study.

## 2 Data

This study is based on measurements of SWIs in water vapour, as well as meteorological variables from the Antarctic Circumnavigation Expedition (ACE), and on a simulation using the isotope-enabled regional numerical weather prediction model COSMO$_{\text{iso}}$ (Pfahl et al., 2012). These two data sets are briefly introduced in the following.

### 2.1 Ship measurement

ACE took place from 21 December 2016 to 19 March 2017 in the Southern Ocean and a variety of meteorological data are available from this expedition (Schmale et al., 2019). In this study, we use the laser spectrometer measurements of SWIs in water vapour and specific humidity ($q_a$) at 13.5 m a.s.l. (Thurnherr et al., 2020), merged sea surface temperature (SST) from in situ measurements and satellite products (Haumann et al., 2020), air temperature ($T_a$) and air pressure ($p_a$) from the onboard weather station (Landwehr et al., 2019), as well as rainfall rates along the ship track from continuous micro rain radar measurements (Gehring et al., 2020). The relative humidity with respect to sea surface temperature $h_s = \frac{q_a}{q_s(SST,p_a)}$ is calculated using the measured $q_a$, SST and $p_a$, where $q_s$ is the saturation specific humidity. All measurement data are used at an hourly time resolution.

### 2.2 Regional numerical model simulation and air parcel trajectories

In this study, SWI signals in the warm sector of an extratropical cyclone will be investigated in a case study using 1-hourly fields from a COSMO$_{\text{iso}}$ simulation in the Southern Indian Ocean. The one-month, nudged COSMO$_{\text{iso}}$ simulation was performed for the time period 13 Dec 2016 to 12 Jan 2017 with a horizontal grid spacing of 0.125°, corresponding to ~14 km, 40 vertical levels and treating deep convection explicitly. The model domain spans an area of 50°x50° and is centred at 47°S, 18°E (Fig. 1a). Six-hourly ECHAM5-wiso fields (Werner et al., 2011) are used as initial and boundary data. Horizontal winds above 850 hPa are spectrally nudged to the ECHAM5-wiso wind field during the COSMO$_{\text{iso}}$ simulation. A detailed description and validation of the simulation was done by Thurnherr et al. (2021).

Using the COSMO$_{\text{iso}}$ three dimensional wind fields, 4-day backward and 2-day forward trajectories are calculated using the Lagrangian analysis tool LAGRANTO (Wernli and Davies, 1997; Sprenger and Wernli, 2015). The trajectories are started at 22 UTC on 26 December 2016 from the warm sector of an extratropical cyclone, more specifically from a region with warm



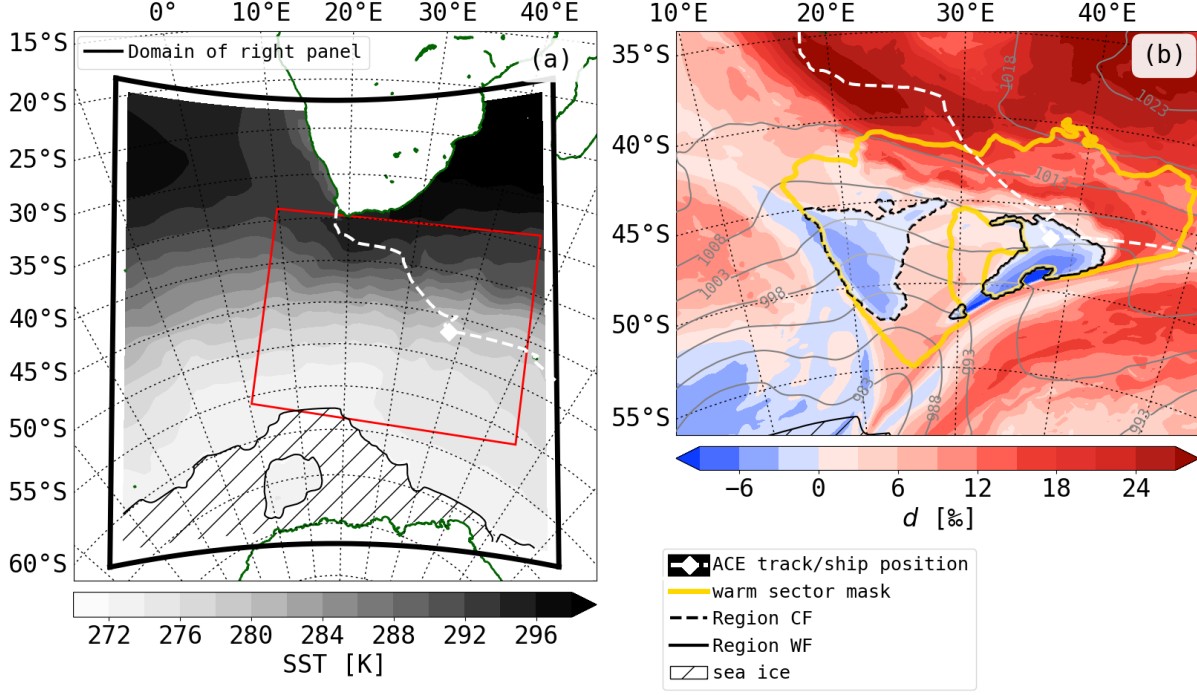

**Figure 1. (a)** Mean SST during the COSMO$_{iso}$ simulation; The model domain is shown by the thick black line and the domain of panel **(b)** with a red rectangle. **(b)** Simulated $d$ on the lowest model level (colours) and sea level pressure at 22 UTC on 26 December 2016 (grey contours, every 5 hPa). Additionally, important regions for the case study are highlighted: the warm sector (yellow line), the region of low d behind the warm front (solid black line, region WF) and ahead of the cold front (dashed black line, region CF) and the ACE track in the model domain (white dashed line; white diamond for the ship position at 22 UTC on 26 December 2016).

temperature advection ($T_a$-SST>1°C) and negative $d$ on the lowest model level (regions CF and WF in Fig. 1b, for details see Sec. 4.2), from every fifth model grid point (i.e. approximately every 70 km) and at heights between 10 and 50 m a.s.l. in 10 m steps. This leads to a total of 2582 trajectories. Several variables are interpolated along the trajectories to analyse the SWI evolution during transport.

## 3 Single-process air parcel models

To quantify the effect of different processes on the SWI signals in the MBL, in particular on observed values of negative $d$ in water vapour in a cyclone's warm sector, single-process models are developed and applied in this study. These models represent the SWI evolution in an air parcel due to three processes separately, which, as discussed in the introduction, are potentially involved in the formation of negative $d$ in the warm sector of cyclones: (i) ocean evaporation during advection over an SST gradient, (ii) dew deposition on the ocean surface, and (iii) cloud formation during a moist adiabatic ascent. The single-process models are set up with a prescribed SST gradient and the moist adiabatic ascent (Rayleigh fractionation) along their path. The





change in the isotopic composition of the air parcel is expressed explicitly as a function of $q_a$ and implicitly as a function of time, due to changes in $q_a$ with time related to surface fluxes or condensation in clouds. In all three models, and in accordance

with the COSMO$_{iso}$ simulation, the formulation by Horita and Wesolowski (1994) is used for the equilibrium fractionation factors and a wind speed independent formulation by Pfahl and Wernli (2009) for the non-equilibrium fractionation factors. For the isotopic composition of the ocean we assume $\delta^{18}O=1\%_o$ and $\delta^2H=1\%_o$, corresponding to the ocean values in COSMO$_{iso}$. In the following subsections, the three single-process air parcel models are introduced and example simulations are discussed. The evolution of the isotope signals is shown in the $q$-$\delta$ and $q$-$d$ phase spaces, which are frequently used in the literature (e.g.

Noone et al., 2011; Benetti et al., 2018).

### 3.1 Moisture uptake by ocean evaporation during advection

Ship-based measurements of SWIs in water vapour are strongly influenced by ocean evaporation. Air exposed to intense ocean evaporation, can be identified by high $d$ in water vapour (e.g. Pfahl and Wernli, 2008; Aemisegger and Sjolte, 2018; Thurnherr et al., 2021) and in downstream precipitation (Aemisegger, 2018). The strength of ocean evaporation depends primarily on

the relative humidity with respect to sea surface temperature, $h_s$, and the wind speed. The lower $h_s$, the stronger the ocean evaporation flux and the stronger the non-equilibrium fractionation effect becomes. For an air parcel experiencing continuous moistening due to ocean evaporation, $h_s$ increases if entrainment of drier air from above is neglected. This leads to a decrease in the ocean evaporation flux and thus a decrease in the strength of non-equilibrium fractionation. A further factor influencing $h_s$ is SST, which, over strong spatial SST gradients, influences the saturation specific humidity $q_s$. Advection in the cold and

warm sectors of extratropical cyclones typically leads to a change in SST seen by the air parcel during transport and thus a change in $h_s$. The evolution of the isotopic composition of an air parcel, which is continuously moistened by ocean evaporation, while it crosses an SST gradient from the warmer to the cooler side is simulated with a single-process air parcel model, referred to as $APM_{evap}$ in the following.

### 155  3.1.1 APM$_{evap}$ setup

Here, we are interested in the SWI evolution along a trajectory in the warm sector moving poleward over the Southern Ocean, over a negative SST gradient during which $d$ is expected to decrease. To simulate the air parcel's movement over an SST gradient a linear decrease in SST from an initial SST$_0$ to a final SST$_{end}$ is predefined in this model. The SST along the trajectories defines the evolution of $q_s$ during the simulation. The initial specific humidity $q_a$ of the air parcel is prescribed

such that the initial $h_s <1.0$. During the simulation, $q_a$ is increased stepwise by an amount $\Delta q$, which is proportional to the evaporation flux $E$, which in turn is proportional to $(q_s-q_a)$, assuming that turbulent mixing is constant. Thus, any mixing with moisture around the parcel is neglected except for the input from ocean evaporation. The evolution of the isotopic composition of the air parcel is calculated iteratively by linear mixing of the air parcel's humidity with the freshly evaporated moisture $\Delta q$.





This gives the isotopic ratio of the water vapour in the air parcel at step $i$+1:

$$R_{\mathrm{a},i+1} = \frac{R_{\mathrm{a},i} \cdot q_{\mathrm{a},i} + R_{\mathrm{flux}} \cdot \Delta q}{q_{\mathrm{a},i+1}}, \tag{1}$$

where $R_{\mathrm{a},i}$ is the isotopic ratio of the water vapour in the air parcel at step $i$ and $q_{\mathrm{a},i+1} = q_{\mathrm{a},i} + \Delta q$. This stepwise increase in $q$ continues until $q_a \approx q_s$, which corresponds to saturated conditions. The isotopic ratio of the evaporation flux $R_{\mathrm{flux}}$ is calculated using the model by Craig and Gordon (1965), with the isotopic ratio of evaporated water vapour defined as

$$R_{\mathrm{flux}} = \frac{\alpha_k \cdot (R_{\mathrm{a}} \cdot h_s - R_{\mathrm{oc}} \cdot \alpha_e)}{h_s - 1}, \tag{2}$$

where $R_{\mathrm{oc}}$ is the isotopic ratio of the ocean surface water. In this air parcel model, the effects of evaporative cooling on the SST and evaporative enrichment on $R_{\mathrm{oc}}$ are neglected.

### 3.1.2 An example simulation of APM$_{\mathrm{evap}}$

For an exemplary APM$_{\mathrm{evap}}$ simulation, the SST gradient is defined by SST$_0$=14°C and a linear decrease of 0.12°C h$^{-1}$, which represents the SST evolution expected along a trajectory moving polewards at a speed of 30 km h$^{-1}$ in the warm sector of an extratropical cyclone. This corresponds to an air parcel moving over a distance of 1000 km within 36 h (see supplement Fig. S3). The simulation is initialised with $q_{\mathrm{a},0}$=5 g kg$^{-1}$ (and, thus, $h_s$=0.5 because $q_s$=10.0 g kg$^{-1}$ at 14°C), $\Delta q = 10^{-3} \cdot (q_s - q_a)$, $\delta^2\mathrm{H}_{\mathrm{a},0}$=−137‰ and $\delta^{18}\mathrm{O}_{\mathrm{a},0}$=−19.5‰, and the results are shown in Fig. 2. The simulation was stopped when $h_s$=90% because ocean evaporation gets very weak close to saturation. Next to the isotopic composition of the water vapour and the evaporation flux, the equilibrium vapour from the ocean surface water is shown in Fig. 2 (orange line), which represents the isotopic composition that the water vapour would have if it was in thermodynamic equilibrium with the ocean surface. The equilibrium vapour composition is equal to $R_{\mathrm{oceq}} = \alpha_e(\mathrm{SST}) \cdot R_{\mathrm{oc}}$ and changes slightly during the simulation due to changes in SST. During the simulation, $\delta^{18}\mathrm{O}_{\mathrm{a}}$ and $\delta^2\mathrm{H}_{\mathrm{a}}$ in the vapour phase increase towards $\delta^{18}\mathrm{O}_{\mathrm{oceq}}$ and $\delta^2\mathrm{H}_{\mathrm{oceq}}$, respectively (Fig. 2a,b). $d_{\mathrm{a}}$ decreases continuously from its initial value of 19.0‰ to 7.6‰ at $h_s$=90% (Fig. 2c). This SWI evolution is the result of the equilibration of the water vapour with the ocean surface, which leads to a general increase in $\delta$-values and decrease in $d$ during the simulation, and the effect of non-equilibrium fractionation, which mostly influence the SWI evolution at low $h_s$. The effect of non-equilibrium fractionation on the SWI evolution during the simulation can be quantified by comparing the simulation with non-equilibrium fractionation ($\alpha_k$<1.0) to a simulation with pure equilibrium fractionation ($\alpha_k$=1.0). $\delta^{18}\mathrm{O}_{\mathrm{flux}}$ and $\delta^2\mathrm{H}_{\mathrm{flux}}$ show lower values in the simulation with non-equilibrium fractionation than during pure equilibrium fractionation, because of the lower diffusion velocity of heavy water molecules compared to the light water molecule. Due to the strong non-equilibrium fractionation, $d$ stays at relatively high values in the beginning of the simulation with $\alpha_k$<1.0. As $h_s$ increases and the effect of non-equilibrium fractionation weakens, $\delta^{18}\mathrm{O}_{\mathrm{a}}$ and $\delta^2\mathrm{H}_{\mathrm{a}}$ increase further and $d$ decreases. The decrease in SST during the simulation leads to a faster saturation of the air parcel and weaker non-equilibrium fractionation, than in the simulation without SST changes (not shown). Therefore, the advection over a negatives SST-gradient accelerates the $d_{\mathrm{a}}$ decrease.

The example simulation with APM$_{\mathrm{evap}}$ shows that continuous ocean evaporation into an air parcel, which is moving over a

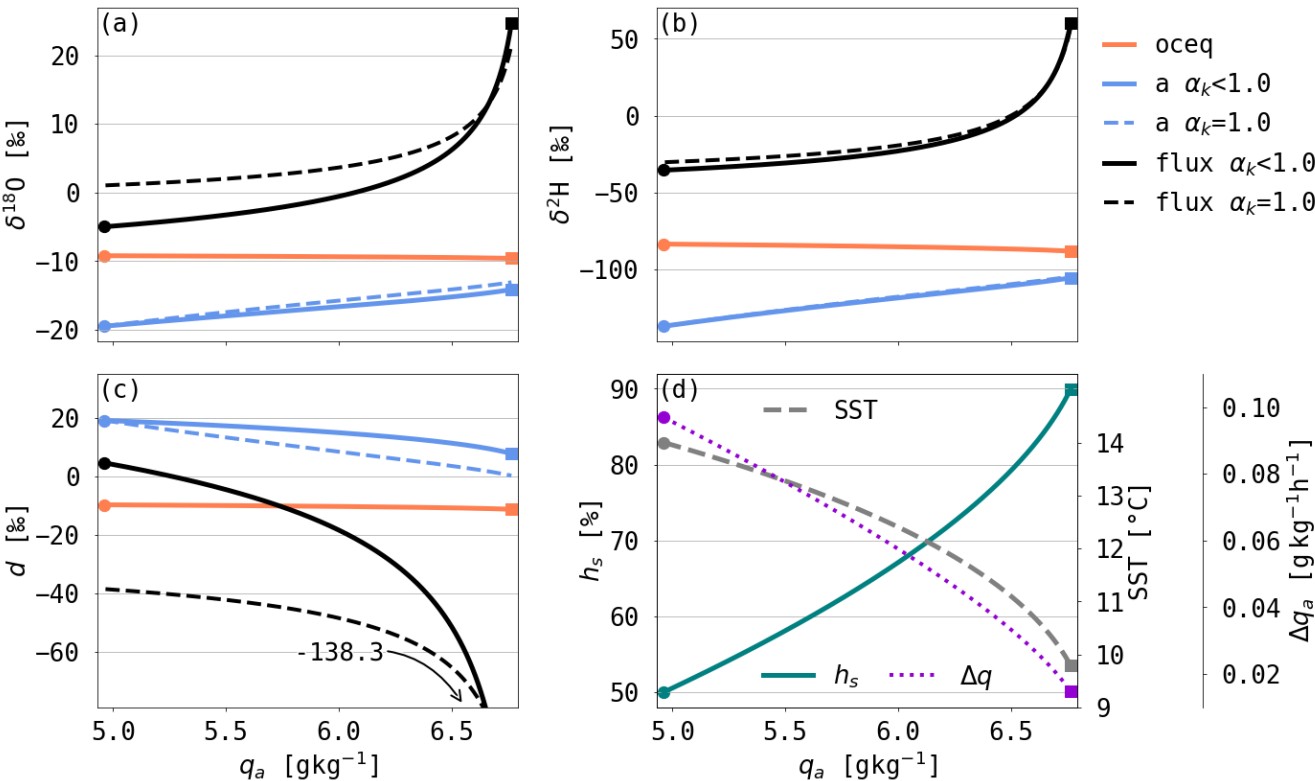

**Figure 2.** Evolution of **(a)** $\delta^{18}O$, **(b)** $\delta^2H$ and **(c)** $d$ as a function of the air parcel's specific humidity $q_a$ in an example simulation with $APM_{evap}$ (see text for details). The blue line shows the isotopic composition of the water vapour in the air parcel [a], the orange line of the equilibrium vapour with the ocean surface [oceq] and the black line of the evaporation flux [flux]. For $d_{flux}$, the end composition is not shown, but indicated by $-138.3‰$ in **(c)**. **(d)** shows SST (grey dashed line), $h_s$ (green solid line) and $\Delta q_a$ (violet dotted line) during the simulation. The values of $\Delta q_a$ are calculated assuming that the SST decrease occurred over 36 h. The simulation is shown with non-equilibrium fractionation ($\alpha_k < 1.0$, solid lines in a-c) and without ($\alpha_k = 1.0$, dashed lines in a-c). The dots denote the start and the squares the end of the simulation.





negative SST gradient, leads to a decrease in $d_a$ and an increase of $\delta^{18}O_a$ and $\delta^2H_a$. The detailed evolution of the water vapour isotope signals depends on the strength of non-equilibrium fractionation, which is as a function of $h_s$ and SST. In Sect. 4, the evolution of SWIs in a simulation with $APM_{evap}$ will be compared to the evolution along $COSMO_{iso}$ backward trajectories, which arrive in the warm sector of an extratropical cyclone and experience ocean evaporation over decreasing SST during

several days before arrival.

The example $APM_{evap}$ simulation discussed here was stopped at $h_s$=90%. If the simulation was not stopped, $h_s$ would eventually exceed 100% if the SST continued to decrease and, thus, the air parcel become oversaturated with respect to the SST. The effect of oversaturated conditions on the SWI composition of water vapour in the air parcel is discussed in the next section with the dew deposition APM.


### 3.2 Moisture loss by dew deposition on the ocean surface

Measurements of SWIs during ACE showed low or negative $d_a$ in combination with high $\delta^2H_a$ and $\delta^{18}O_a$ during periods of warm temperature advection, i.e. $T_a-SST>1°C$ and positive anomalies in $q_a$ (Thurnherr et al., 2021). In these situations, the atmosphere is often oversaturated with respect to the SST such that dew deposition occurs on the ocean surface. To model the

corresponding evolution of the SWI composition of the atmospheric water vapour in an oversaturated air parcel moving over a negative SST gradient and arriving in the warm sector of an extratropical cyclone an approach similar to $APM_{evap}$ is chosen. In the following, the model setup of the dew deposition air parcel model ($APM_{dew}$) is described and the SWI evolution during an exemplary simulation is shown and discussed.

#### 3.2.1 $APM_{dew}$ setup

The $APM_{dew}$ describes the evolution of $q_a$ and SWIs in an air parcel, which experiences continuous dew deposition while moving over an SST gradient. A linear decrease in SST during the simulation is assumed to represent the poleward movement of the air parcel over the Southern Ocean in the same way as in $APM_{evap}$. In contrast to $APM_{evap}$, there is only a moderate decrease in SST in $APM_{dew}$ based on the SST gradient along trajectories experiencing dew deposition in the $COSMO_{iso}$ domain

of this study.

We initiate an air parcel with a specific humidity $q_{a,0} > q_{s,0}$ and, thus, a supersaturation of $h_{s,0} = \frac{q_{a,0}}{q_{s,0}}$. This air parcel equilibrates with the ocean surface during the simulation by iteratively decreasing $q_a$ in the air parcel by an amount $\Delta q$, which is proportional to $(q_a-q_s)$, equivalent to the approach in $APM_{evap}$. The isotopic composition of the water vapor can be iteratively calculated which leads to the following expression of $R_a$ at simulation step $i+1$:

$$R_{a,i+1} = \frac{R_{a,i} \cdot q_{a,i} - R_{flux} \cdot \Delta q}{q_{a,i+1}} \qquad (3)$$

$R_{flux}$ is equivalent to the composition of the evaporation flux given by Craig and Gordon (1965) (Eq. 2) as isotopic fractionation during dew deposition can be described by same bulk aerodynamic formulation as ocean evaporation (see Appendix A1).





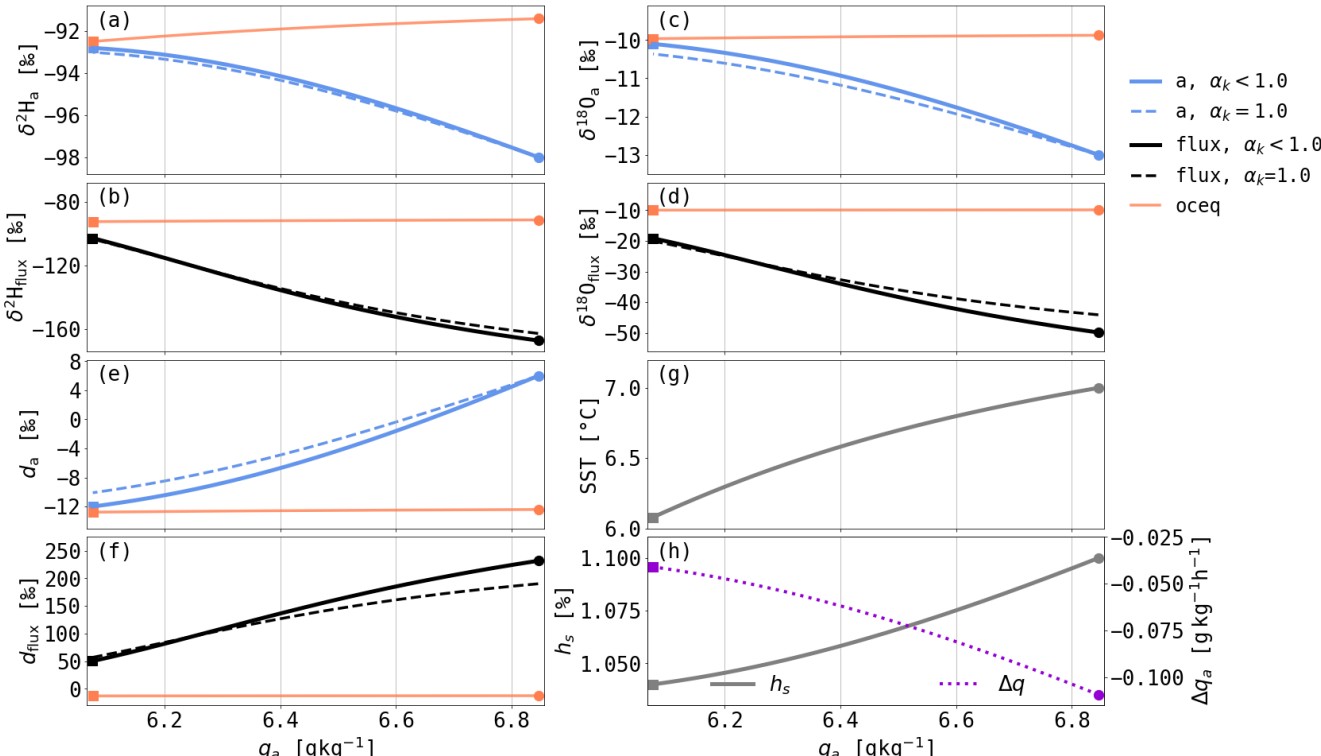

**Figure 3.** Evolution of **(a,b)** $\delta^2H$, **(c,d)** $\delta^{18}O$ and **(e,f)** $d$ as a function of the air parcel's specific humidity $q_a$ in an example simulation with $APM_{dew}$ (see text for details). The blue line shows the isotopic composition of the water vapour in the air parcel [a], the orange line of the equilibrium vapour with the ocean surface [oceq] and the black line of the dew deposition flux [flux]. The simulation is shown with non-equilibrium fractionation ($\alpha_k$ <1.0, solid lines in a-f) and without ($\alpha_k$=1.0, dashed lines in a-f). Furthermore, the evolution of **(g)** SST and **(h)** $h_s$ (grey solid line) and $\Delta q_a$ (purple dashed line) during the simulations is shown. The values of $\Delta q$ are calculated assuming that the SST decrease of 0.9°C occurred over 12 h. Notice, the simulations start to the right (dots) and end to the left (square) as the air parcel looses moisture during dew deposition.

Furthermore, it is assumed that the $R_{oc}$ is not altered by the small amount of deposited dew and that temperature variations due to equilibration or phase changes can be neglected. In the following, a simulation of the $APM_{dew}$ is described in detail to
highlight the key factors influencing the SWI evolution during dew deposition.

### 3.2.2    An example simulation with $APM_{dew}$

Figure 3 shows the evolution of the isotopic composition of the water vapour and the dew deposition flux as a function of $q_a$ for a simulation using the $APM_{dew}$. During the simulation, SST decreases from 7°C (thus $q_{s,0}$=6.2 g kg$^{-1}$) to 6.1°C. These values are chosen to represent typical environmental conditions for an air parcel experiencing dew deposition for 12 h in the
warm sector of an extratropical cyclone (see supplement Fig. S3). The simulation is initialised with $h_s$=1.1, which means that





$q_{a,0}$=6.8 g kg$-1$, $\Delta q$=8·10$^{-4}$·$(q_s - q_a)$, $\delta^2$H$_{a,0}$=$-98$‰, and $\delta^{18}$O$_{a,0}$=$-13$‰. The simulation is stopped if $h_s$ decreases below 1.04 because the moisture fluxes become very small close to equilibrium conditions.

The APM$_{dew}$ simulation shows a decrease in $d_a$ while $\delta^{18}$O$_a$ and $\delta^2$H$_a$ increase toward $\delta^{18}$O$_{oceq}$ and $\delta^2$H$_{oceq}$, respectively. The decrease of $d_a$ of 18.0‰ is in a similar range as the decrease in $d_a$ during the example simulation with APM$_{evap}$ (Fig. 2).

Non-equilibrium fractionation has an opposite effect on the changes in $\delta^{18}$O$_a$, $\delta^2$H$_a$ and $d_a$ in APM$_{dew}$ compared to APM$_{evap}$, because of the opposite direction of the moisture flux that is here from the atmosphere to the ocean.

The specific evolution during the simulations depends on the one hand on the initial isotopic composition of the air parcel, which affects the rate of change in $\delta$-values during the simulation due to its control on the ocean-atmosphere isotopic gradient. On the other hand, the initial $h_s$ sets the strength of the increase in $\delta$-values during the simulation as it constrains the strength

of the effects from non-equilibrium fractionation. Furthermore, the evolution of SST and $q_a$ may change depending on the SST gradient and the speed of the air parcels. If the initial values correspond to typical environmental conditions in the warm sector, a decrease in $d_a$ and increase in $\delta^{18}$O$_a$ and $\delta^2$H$_a$ is observed in all simulations. The performance of the APM$_{dew}$ compared to the SWI evolution along COSMO$_{iso}$ trajectories will be further analysed in Sect. 4.3.

### 3.3 Rayleigh fractionation during moist adiabatic ascent

For air parcels close to the ocean-atmosphere interface, ocean evaporation and dew deposition are expected to be more important for the isotopic composition of water vapour than moist processes at more elevated altitudes. Nonetheless, SWIs in the water vapour might carry a signal from more distant processes than seen close to the atmosphere-ocean interface. Trajectories arriving in the warm sector might have been subject to cloud formation during which a decrease in $d$ occurs. Therefore, the effect of Rayleigh fractionation during a moist adiabatic ascent of an air parcel is modelled with APM$_{ray}$.

During a moist adiabatic ascent, water vapour condenses in the ascending air parcel, which leads to the formation of cloud droplets and changes the isotopic composition of the remaining atmospheric water vapour. The SWI evolution in the air parcel during this process can be modelled using a Rayleigh fractionation model based on the temperature and moisture evolution of the moist-adiabatic ascent. For simplification, it is here assumed, that the condensed cloud droplets do not interact with the surrounding water vapour after condensation and are immediately removed. This provides a first order understanding of isotopic

variations in water vapour along trajectories during a moist adiabatic ascent. In APM$_{ray}$, the temperature and humidity evolution is approximated by the evolution during a moist adiabatic ascent starting with $T_{a,0}$ and $p_0$ and is modelled in a similar way as in Dütsch et al. (2017). The isotopic ratio of the water vapour during Rayleigh fractionation at step $i+1$ is defined iteratively by $R_{a,i+1} = f^{(\alpha_{eff}-1)} \cdot R_{a,i}$, where $f = \frac{q_{a,i+1}}{q_{a,i}}$ is the remaining fraction of moisture in the vapour phase. The change in $q_a$ from $i$ to $i+1$ corresponds to the excess moisture at step $i$ relative to $q_s$. $R_{a,i}$ is the isotopic ratio of the water vapour in the air

parcel at step $i$. $\alpha_{eff} > 1$ is the isotopic fractionation factor of the water vapour isotope signals which depends on temperature at step i and the share of liquid and solid condensate during the moist-adiabatic ascent. The equilibrium fractionation factor in vapour with respect to ice from Majoube (1971) is used and adjusted for supersaturation over ice (Jouzel and Merlivat, 1984). A detailed description of the model can be found in Thurnherr et al. (2021).

An example simulation of APM$_{ray}$ initiated with $T_{a,0}$=8°C (which gives $q_{a,0}$=6.7 $g\,kg^{-1}$), $\Delta$SST=1°C, $\delta^{18}$O$_{a,0} = -15.0$‰



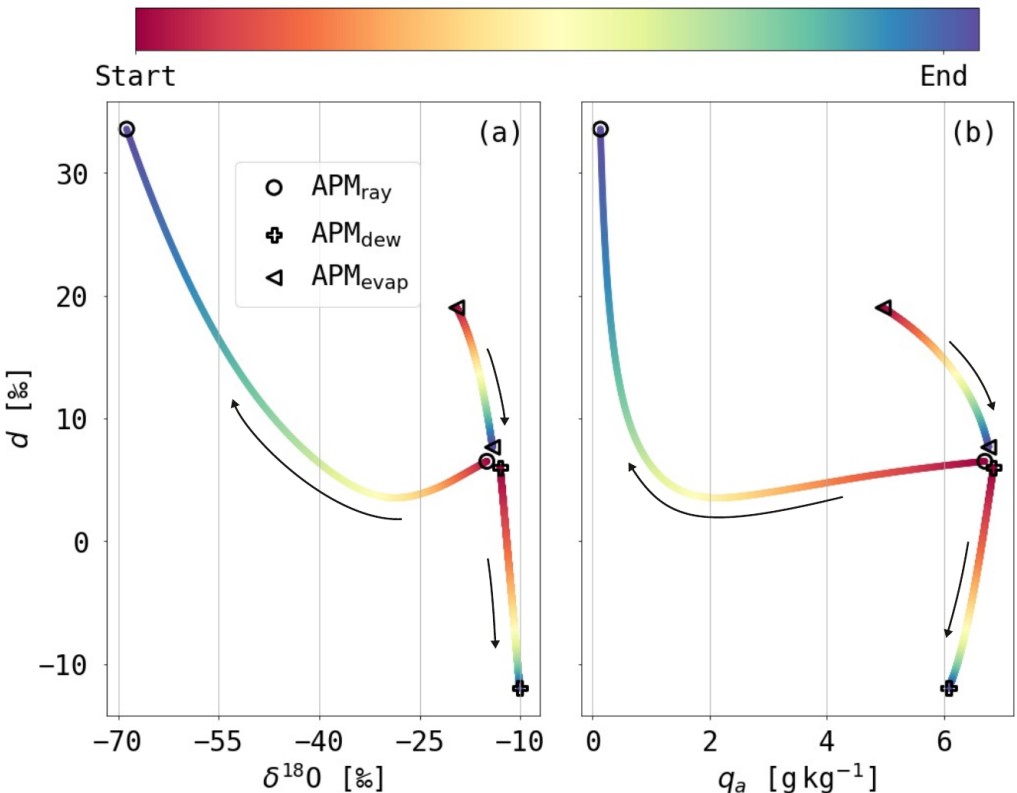

**Figure 4.** Phase space diagrams of **(a)** $d$ versus $\delta^{18}O$ and **(b)** $d$ versus $q_a$ for the example simulations with $APM_{evap}$ (◁), $APM_{dew}$ (+) and $APM_{ray}$ (o) discussed in Sect. 3.1, 3.2 and 3.3. The lines are coloured from the start to the end position. The black arrows show the direction from the start to the end during the simulation.

and $\delta^2H_{a,0} = -98‰$ is shown in Fig. 4. A decrease in $d_a$ can be seen in simulations with the $APM_{ray}$ due to the decrease in $T_a$ during the ascent until less than 25% of the initial moisture is left in the air parcel. Once, the amount of heavy isotopes in the vapour phase becomes very small, $d_a$ increases by definition (due to the non-linearity of the $\delta$-scale, see also Dütsch et al., 2017) for all scenarios. This example illustrates that the condensation of cloud water is a process that can lead to a decrease in $d_a$.

All three air parcel models introduced here show a decrease in $d_a$, but different evolutions of $q_a$ and $\delta^{18}O_a$. This different behaviour of the three air parcel models can be summarised in the $d_a$-$\delta^{18}O_a$- and $d_a$-$q_a$-phase space as discussed in the next section.



### 3.4 Comparison of the single-process air parcel models

The effects of the $APM_{evap}$, $APM_{dew}$ and $APM_{ray}$ on the isotopic composition of water vapour can be separated using phase space diagrams (Fig. 4). In a $d_a$-$\delta^{18}O_a$-phase space (Fig. 4a), each model shows a specific evolution during the simulation. The simulations from all APMs show a decrease in $d_a$ during the first part of their evolution, but they show a different behaviour with respect to $\delta^{18}O_a$ and $q_a$. $\delta^{18}O_a$ increases during the simulations with $APM_{dew}$ and $APM_{evap}$, while it decreases continuously in the simulation with $APM_{ray}$. This leads to a negative slope for $APM_{dew}$ and $APM_{evap}$ and a positive slope for $APM_{ray}$. The

decrease in $d_a$ for $APM_{evap}$ can be separated from the $d_a$-decrease for $APM_{dew}$ and $APM_{ray}$ in the $d_a$-$q_a$-phase space. While $q_a$ decreases in the $APM_{ray}$ and $APM_{dew}$ simulations, $q_a$ increases in the $APM_{evap}$ simulation. Therefore, in the $d_a$-$\delta^{18}O_a$- and the $d_a$-$q_a$-phase spaces, each APM has a unique signature. These different signatures in the phase spaces will be used in Sect. 4.3 to disentangle the importance of these different moist processes along $COSMO_{iso}$ trajectories in the warm sector of an extratropical cyclone.

## 4 Case study of warm temperature advection in the Southern Indian Ocean

The effect of ocean evaporation over an SST gradient, dew deposition, and cloud formation on the isotopic composition of the near-surface water vapour is analysed for a case of warm temperature advection in the warm sector of a Southern Ocean midlatitude cyclone. The focus on a specific case study enables a detailed analysis of processes influencing the isotopic composition of the warm sector and for an evaluation of the APMs. The case study cyclone occurred during ACE from 24 to 28 December

2016 South of Marion Island in the South Indian Ocean, which allows for an assessment of the $COSMO_{iso}$ performance by comparing measured and modelled data. First, we give an overview of the synoptic situation and compare measurements with the $COSMO_{iso}$ simulation for this case study. Second, we describe the SWI evolution of the warm sector and, third, we apply the APMs to assess how well they can simulate the SWI evolution along $COSMO_{iso}$ trajectories.

### 4.1 Synoptic overview

The warm sector of an extratropical cyclone passes south of Marion Island (black dot in Fig. 5b-d) from 24 to 28 December 2016. At 22 UTC on 25 December 2016, this cyclone is positioned at 5°W and 55°S. While the cyclone center is already uniformly cold, it is associated with prominent warm and cold fronts that meet near 20°E/50°S. Both fronts are visible by clear kinks in the sea level pressure contours and sharp gradients in THE. The focus of this study is the wide warm sector spanned by

these fronts. An elongated band of high $\theta_e$ (see Fig. 5a,b) leads to the highest $\theta_e$ values behind the warm front between 22 UTC on 25 and 26 December 2016. A second area of high $\theta_e$ is located ahead of the cold front (see Fig. 5b,c). These two regions of high $\theta_e$ are separated by a weak secondary warm front at 22 UTC on 26 December 2016 associated with a short-lived meso-cyclone forming and dissolving on 25 December (see weak low pressure system at 52° S and 18° E in Fig. 5b). The strongest precipitation is seen along the cold and warm front and, towards the end of the case study, along the occluded front, which

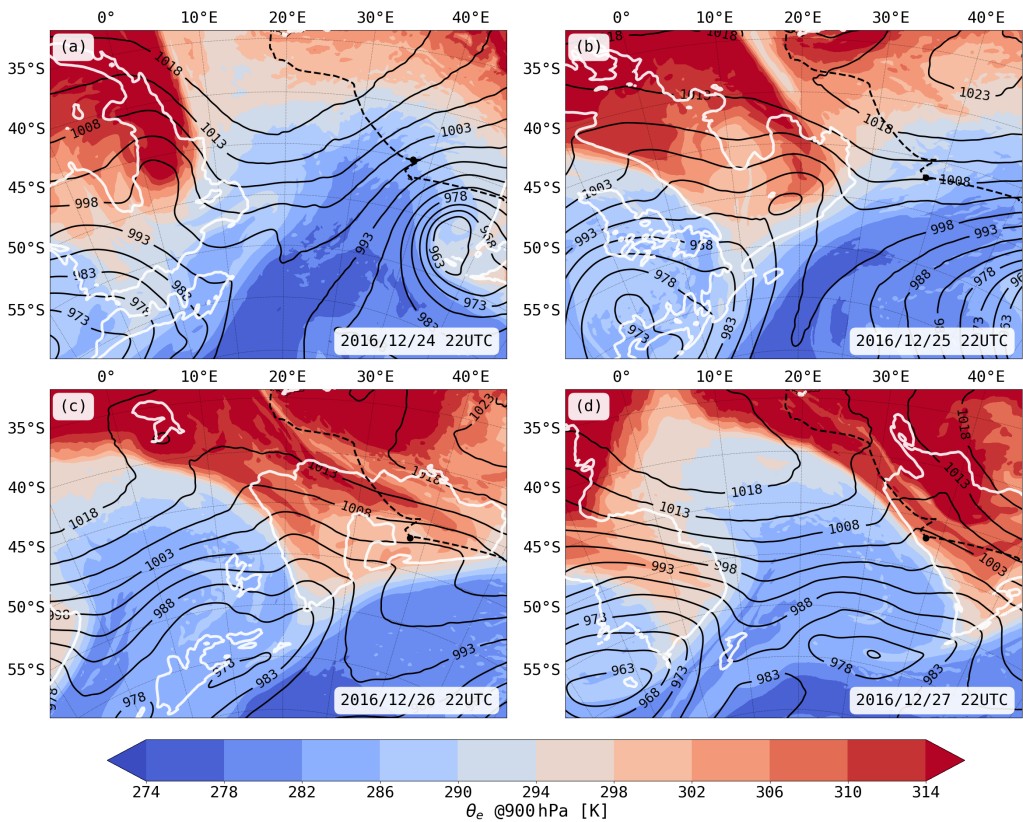

**Figure 5.** Horizontal cross sections of equivalent potential temperature at 900 hPa ($\Theta_e$ @900 hPa) and sea level pressure (black contours, in hPa) at four time steps during the warm advection case study. The black dashed line shows the ACE ship track and the black dot denotes the position of the research vessel. The white contours show that warm temperature advection mask.

forms around 25°E and 50°S on 26 December 2016 (see supplement Fig. S1c). Weak, shallow-convective precipitation occurs in the cold sector. The region of the warm sector is defined by the near-surface air-sea temperature difference as described in Thurnherr et al. (2021). If the difference between the 2 m air temperature and the sea surface temperature is above 1°C, the region is associated with the advection of warm and moist air. The warm sector region is indicated by the white contours in Fig. 5 and encompasses a triangular region in between the cold and warm front. When referring to the warm sector in the following,

we are referring to this area of warm temperature advection.

The ACE measurements for this case study were taken at Marion Island, where the research vessel was anchored during the cyclone passage (black dot in Fig. 5). The vessel stayed in the warm sector for 36 h. A good agreement of measured and simulated $h_s$ and $q_a$ can be seen (Fig. 6). The simulated precipitation compares well with the measurements except for the few hours around 00 UTC on 26 December 2016, during which enhanced precipitation is simulated, while no precipitation has

been measured. This overestimation led to too high $h_s$ and $q_a$ in the simulation in this time period. Measured $h_s$ stays above





1.0 for nearly the entire event from 15 UTC on 26 December to 00 UTC on 28 December 2016 (Fig. 6a). Simultaneously, simulated ocean evaporation is negative, indicating dew deposition on the ocean surface (Fig. 6b). Highest dew deposition rates coincide with high $h_s$ and $q_a$ and a very shallow diagnosed boundary layer height in the model as expected in stable near-surface conditions. There are two peaks in dew deposition rates at the measurement site, which correspond to two areas of high $\theta_e$ within the warm sector (Fig. 5). The first area of dew deposition lies behind the warm front, the second ahead of the cold front. In between these two areas, weakly positive ocean evaporation occurs (Fig. 6b). This specific structure of $h_s$ and $\theta_e$ in the warm sector is imprinted in the isotopic composition of the near-surface water vapour. The distribution and evolution of SWIs in the warm sector during the case study is discussed in the next section.

### 4.2 Isotope signals

The measured isotopic composition of water vapour during the cyclone passage shows a decrease of $10\,‰$ in $d$ to $-5.4\,‰$ and an increase in $\delta^{18}O$ of $5\,‰$ and $\delta^2H$ of $30\,‰$ up to $-11\,‰$ and $-88\,‰$, respectively (Fig. 6d-f). The simulated timelines of $d$, $\delta^{18}O$ and $\delta^2H$ agree qualitatively and quantitatively well with the measurements in the warm sector showing a Pearson correlation above 0.7 for all isotope variables and a mean absolute error of $3.6\,‰$, $1.4\,‰$ and $9.6\,‰$, respectively. This good agreement of SWIs in COSMO$_{iso}$ and the ACE measurements allows for further analysis of the SWI evolution in the COSMO$_{iso}$ simulation for this case study. Possible reasons for deviations were discussed in Thurnherr et al. (2021). The choice of a too strong non-equilibrium fractionation factor for ocean evaporation being the most important one.

As shown in Fig.7, the development of $d$ during the cyclone's passage is characterised by negative $d$ close to the cyclone center and along the fronts in the warm sector, clearly visible at 22 UTC on 26 December 2016 (white contours in Fig. 7c). These two Eulerian features are confined to the areas behind the warm front and ahead of the cold front, respectively. In between the low $d$ regions, there is a region of $d$ above $0\,‰$ overlapping with the region of low $\Theta_e$ in the warm sector (Fig. 5c). The low $d$ region behind the warm front (region WF) shows continuously low $d$ below $-9\,‰$ and stays situated in a region of warm advection within the warm sector from Dec 24 to 27. The low $d$ region ahead of the cold front (region CF) is more variable in sign, shape and position than the warm frontal low $d$ region. $d$ decreases in region CF during the four days of the case study, from values mainly above $-6\,‰$ to values below $-9\,‰$. Shortly after the dissolution of the short-lived meso-cyclone, the area of region CF increases from a confined region close to the cold front to a larger triangular region spanning towards the middle of the warm advection mask incorporating the warm sector of the meso-cyclone (compare Fig.7b and c).

Additionally to the evolution of these Eulerian $d$ features during the case study, the position of backward and forward trajectories starting below $50\,m$ a.s.l. from these two low $d$ regions in the warm sector at 22 UTC on 26 December 2016 are shown in Fig. 7 (see also supplement Fig. S3 for the full trajectory paths). By construction, at 22 UTC on 26 December 2016 (Fig.7c) all trajectories lie within the regions CF and WF. The backward trajectories entered the marine boundary layer at latest $84\,h$ before arrival in the low $d$ regions (see supplement Fig. S4d), which they enter only a few hours before arrival. Most strikingly, the trajectories entering region WF resided in a region of very high $d$ above $20\,‰$ a day before arrival (circles in Fig.7b). Therefore, these trajectories experience a strong decrease in $d$ within $24\,h$ before arrival in region WF. Similarly, the trajectories arriving ahead of the cold front enter region CF within the $24\,h$ before arrival. Furthermore, the back-trajectories arriving in region

**Figure 6.** Time evolution of 1-hourly **(a)** $h_s$, **(b)** ocean evaporation $E$ (positive upward) and boundary layer height $BLH$, **(c)** precipitation rate $R$ and mean cloud water content on the five lowest model levels (10-200 m a.s.l.) (QC), **(d)** $d$, **(e)** $\delta^{18}O$, **(f)** $\delta^2H$ from ACE measurements and COSMO$_{iso}$ simulations interpolated along the ship track during the warm temperature advection event at Marion Island from 26 Dec to 28 Dec 2016. The blue lines show the measurements with 1-hourly standard deviation, the black, green and pink lines the simulated values including the variability from 10°-shifts of the ACE track (black, green and pink areas). The vertical red dashed line denotes 22 UTC on 26 December 2016. The shaded grey areas correspond the time period of the warm temperature advection event.



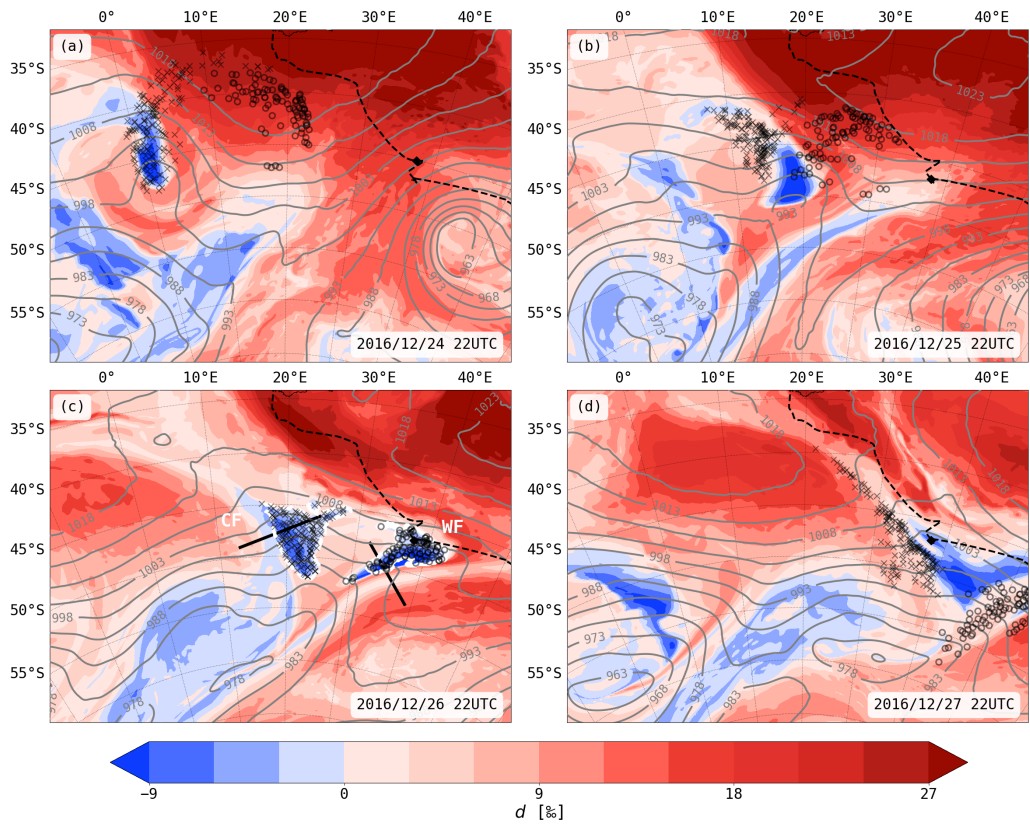

**Figure 7.** Horizontal cross sections of $d$ at the lowest model level and sea level pressure (grey contours, in hPa) at four time steps during the warm advection case study. The black dashed line shows the ACE ship track and the black square denotes the position of the research vessel. The white framed regions in panel (c) show the starting regions WF and CF of the backward and forward trajectories at 22 UTC on 26 December 2016. Circle and crosses show the position of trajectories starting below 50 m a.s.l. in the regions WF and CF, respectively. The black solid lines in (c) denote the positions of the vertical crosssection across the cold and warm front, respectively, in Fig. 8.

CF, were located in region WF 48 h before arrival also coming from a region of high $d$ with values above 20‰ (Fig.7a and supplement Fig. S4). The low $d$ regions in the lower MBL are therefore constantly rebuilt by newly arriving trajectories, which saw a decrease in $d$ just shortly before arrival in the low $d$ regions. This indicates that the low $d$ in these Eulerian features is not materially conserved, but constantly renewed. Fast changes in $d$ are therefore important to keep these low $d$ features in the lower MBL alive and the single-process APMs can help to identify the processes leading to these strong and rapid decreases

in $d$.

  So far, we only discussed the $d$ patterns on the lowest model level at approximately 10 m.a.s.l. A vertical crosssection of $d$ across the cold and warm front at 22 UTC on 26 December 2016 is shown in Fig. 8 to illustrate the vertical extent of these near-surface low $d$ regions. Occurrences of low $d$ behind the warm front lie entirely within the warm advection region and behind the strong temperature gradient across the warm front (Fig 8a). The lowest $d$ values occur in the lower MBL and directly along the warm





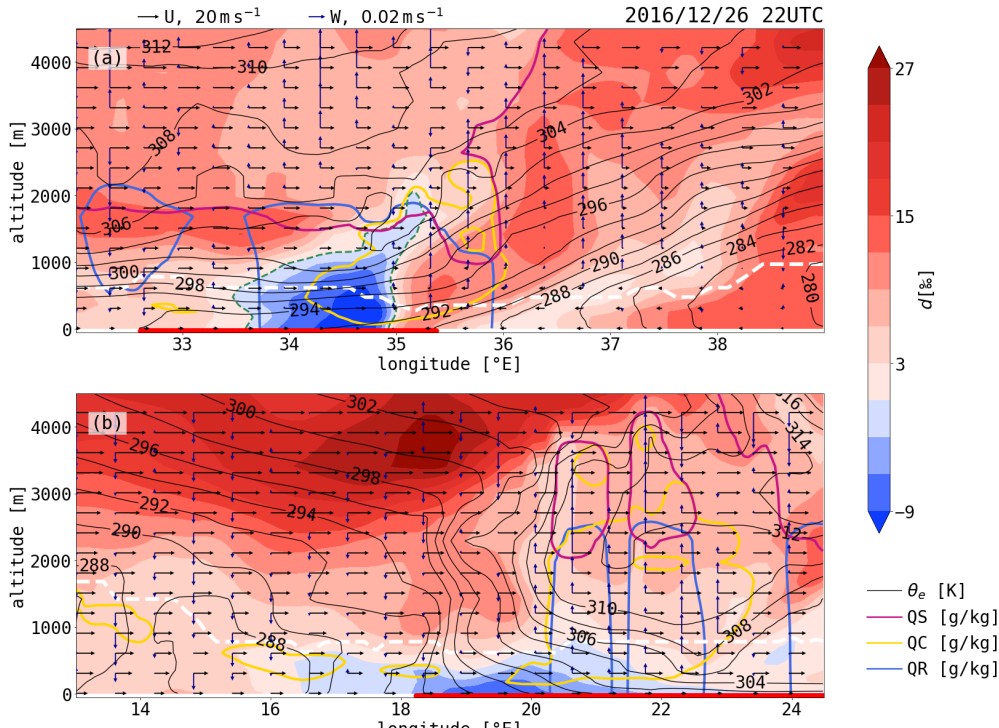

**Figure 8.** Vertical cross sections of $d$ across the **(a)** warm front and **(b)** cold front at 22 UTC on 26 December 2016. The equivalent potential temperature ($\theta_e$, black contours), the snow water content (QS, pink frame), the cloud water content above 0.1 g kg$^{-1}$ (QC, yellow frame), the rain water content above 0.005 g kg$^{-1}$ (QR, blue frame), the boundary layer height (white, dashed line), horizontal (U, black) and vertical (W, blue) wind vectors are shown. The red lines at the bottom of the panels show the regions of warm temperature advection. The orientation of the cross sections is shown in Fig. 7c.

front. In contrast to the cold front, low $d$ is also seen above the MBL. The area of low $d$ might extend to higher altitudes,due to the strong vertical wind shear at lower levels inducing turbulence, despite the stable stratification and leading to vertical mixing (low bulk Richardson numbers can be observed in this region, not shown). In the upper MBL behind the warm front, the low $d$ values might be further transported upward and eastward by large-scale advection. This is supported by the wind patterns in the low $d$ region and highlights the importance of vertical mixing combined with large-scale transport in the frontal region to

shape the $d$ distribution in water vapour. Rain and cloud water are present for most of the low $d$ region behind the warm front. Across the cold front, low $d$ is confined to the lower MBL and covers a region, which mainly overlaps with warm advection, but which also spreads towards the cold side of the front (see negative $d$ west of 18$^o$E in Fig. 8b). The cold front does not show a typical temperature gradient with increasing equivalent potential temperature towards the warm sector and with height. A weak temperature gradient is visible close to surface between 19 an 21$^o$E. An upper level cold front can be observed above

2000 m. Two precipitation cells with strong showers are present at the eastern edge of the low $d$ region, while the lowest $d$ values are located in cloud and rain free areas.





The evolution of $\delta^{18}$O and $\delta^2$H in the warm sector show a pattern opposite to $d$. Due to the similarity of the variability in $\delta^{18}$O and $\delta^2$H only a description of $\delta^{18}$O is given in the following. $\delta^{18}$O shows larger values in the warm sector, with highest values lying in the regions CF and WF (see supplement Fig. S2). A prominent feature are very low $\delta^{18}$O values along the fronts

showing the imprint of frontal precipitation along the warm front (and less strongly along the cold front), most likely due to below-cloud interaction. These low $\delta^{18}$O regions overlap with regions of high $d$.

To understand which processes are most important for these specific isotopic patters, and especially for the low $d$ values seen in the warm sector, the evolution of SWIs in water vapour is analysed using the COSMO$_{\mathrm{iso}}$ trajectories and applying the single-process APMs in the following.


### 4.3 Application of the single-process APMs along the COSMO$_{\mathrm{iso}}$ trajectories

The single-process APMs introduced in Sec. 3 simulate the effect of three processes on the isotopic composition of water vapour in an air parcel during transport: (1) Decreasing $d$ due to ocean evaporation during the movement over a negative SST gradient is simulated with APM$_{\mathrm{evap}}$. (2) If the air parcel becomes oversaturated ($h_s$>1.0), $d$ decreases due to dew deposition,

which is simulated with APM$_{\mathrm{dew}}$. (3) The formation of cloud droplets during a moist-adiabatic ascent of an air parcel can also lead to a decrease in $d$ as simulated with APM$_{\mathrm{ray}}$. In the following the $d$-evolution along COSMO$_{\mathrm{iso}}$ trajectories is compared to these APM simulations to assess if the single-process APMs are able to simulate the specific changes in $d$ seen along the trajectories.

The evolution of SWIs along the trajectories together with the APM evolutions is studied in $d$-$q_a$ and $d$-$\delta^{18}$O phase space dia-

grams (compare Fig. 4). Fig.9a,c shows the evolution of $d$ and $q_a$ along 4 d trajectories arriving in region WF. Four days before arrival, these trajectories were situated in an evaporative environment and experienced an increase in $d$ and $q_a$ simultaneously with increasing ocean evaporation. $d$ peaks on average 34 h before arrival and starts to decrease while $q_a$ is still increasing. On average 18 h before arrival $q_a$ peaks, and starts decreasing thereafter. Shortly before arrival, some trajectories see again an increase in $d$ by a few ‰ to a final $d$ of $-4.8\,[-8.4\text{-}\,-2.0]\,‰$ (the brackets denote the 25 to 75 percentile range). The

trajectories arriving in region CF (Fig.9b,d) show a similar $d$ and $q_a$ evolution in the 4 days before arrival as the ones arriving in region WF. The main differences are a higher maximum in $q_a$ and two local minima in $d$. A first minimum in $d$ is seen, on average, 40 h before arrival when the trajectories pass the low $d$ region behind the warm front in the warm sector of the forming meso-cyclone (compare Figs. 5a and 7a). A second minimum occurs upon arrival when the trajectories enter the low $d$ region ahead of the cold front.

The variations in $d$ and $\delta^{18}$O along the trajectories follows specific patters, which are also seen in idealised APM simulations. The decrease in $d$ while $q_a$ is still increasing follows a similar pattern as seen with the APM$_{\mathrm{evap}}$ (dashed lines in 9c,d). The following decrease in $d$ with a simultaneous decrease in $q_a$ follows a similar pattern as the idealised APM$_{\mathrm{dew}}$ simulations (solid lines in 9c,d). These idealised simulations use a prescribed $q_a$ and SST evolution as introduced in Sec. 3.1 and 3.2. For both APM$_{\mathrm{evap}}$ and APM$_{\mathrm{dew}}$, the idealised simulations show a similar behaviour in the $q_a$-$d$ phase space as the trajectories, but

do not cover the entire range of values simulated along the COSMO$_{\mathrm{iso}}$ trajectories. Especially the lowest $d$ values along the

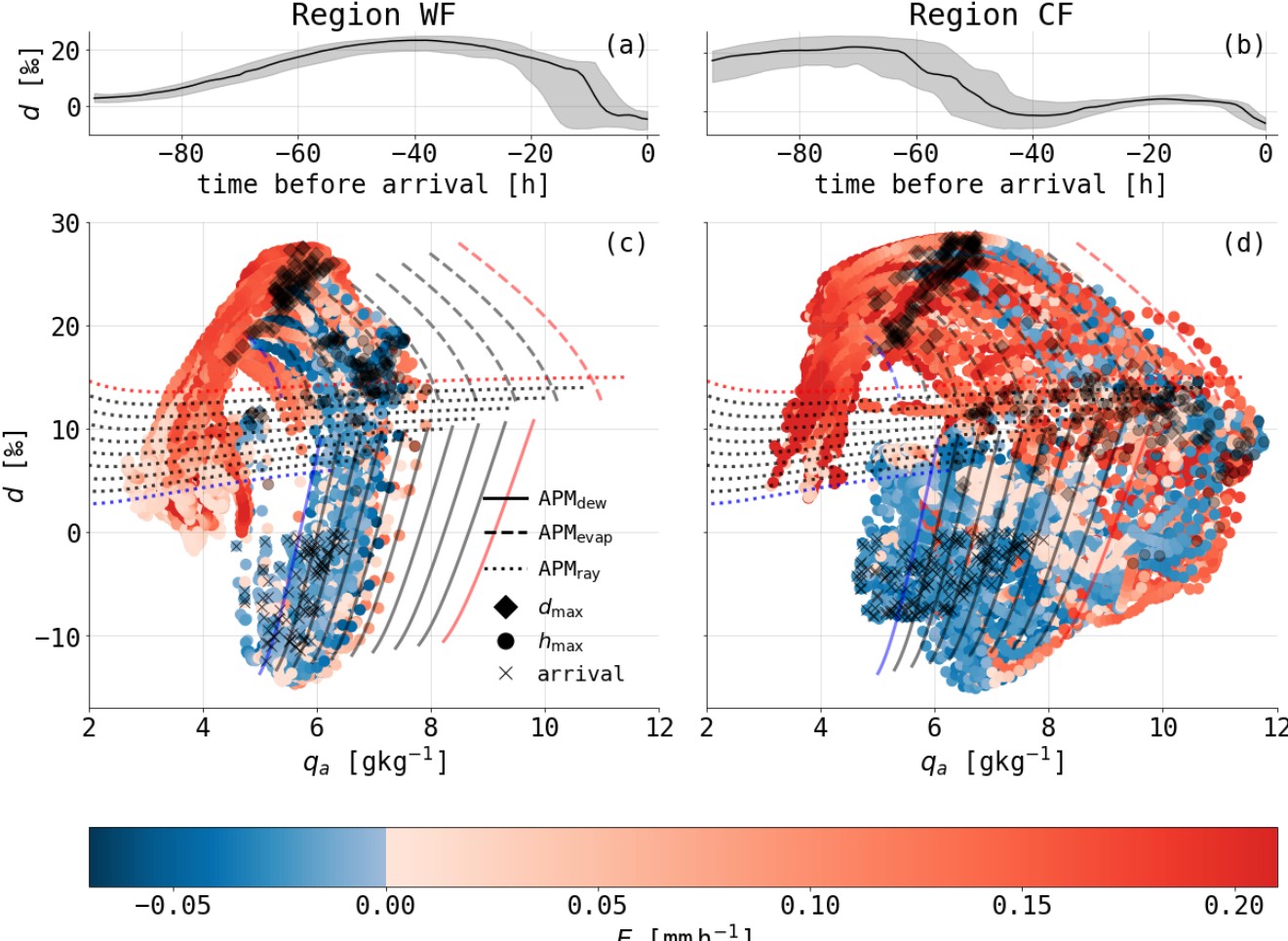

**Figure 9.** Evolution of $d$ (median and [25,75] percentile range) along 4-day backward trajectories starting at 22 UTC on 26 December 2016 below 50 m **(a)** behind the warm front (region WF) and **(b)** ahead of the cold front (region CF). Phase space diagrams of $d$ versus $q_a$ for **(c)** region WF and **(d)** region CF (hourly values along the trajectories are shown by dots, coloured by ocean evaporation (E), which is defined positive upward). The results of the APMs are shown by black lines, solid for $APM_{dew}$, dashed for $APM_{evap}$ and dotted for $APM_{ray}$. The red lines show the simulations with the highest SST ($T_a$ for $APM_{ray}$), the blue lines with the lowest. For details on the model setups, see Appendix Table B1 and text. Markers denote specific events along the trajectories: the time of highest $d$ (black diamond), the time of highest $h_s$ (black dot) and the time of arrival at the measurement site (black cross).

trajectories are not reproduced by the idealised APM simulations. $AMP_{ray}$ (dotted lines in 9c,d) simulations show a too weak decrease in $d$ to explain changes in $d$ and $q_a$ along the trajectories arriving in regions WF and CF.

The assumed, idealised SST and $q_a$ evolution in the idealised APM simulations might lead to the difference between in the APMs and the $COSMO_{iso}$ trajectories. Instead of these idealised assumptions, the evolution of $q_a$ and SST along the $COSMO_{iso}$





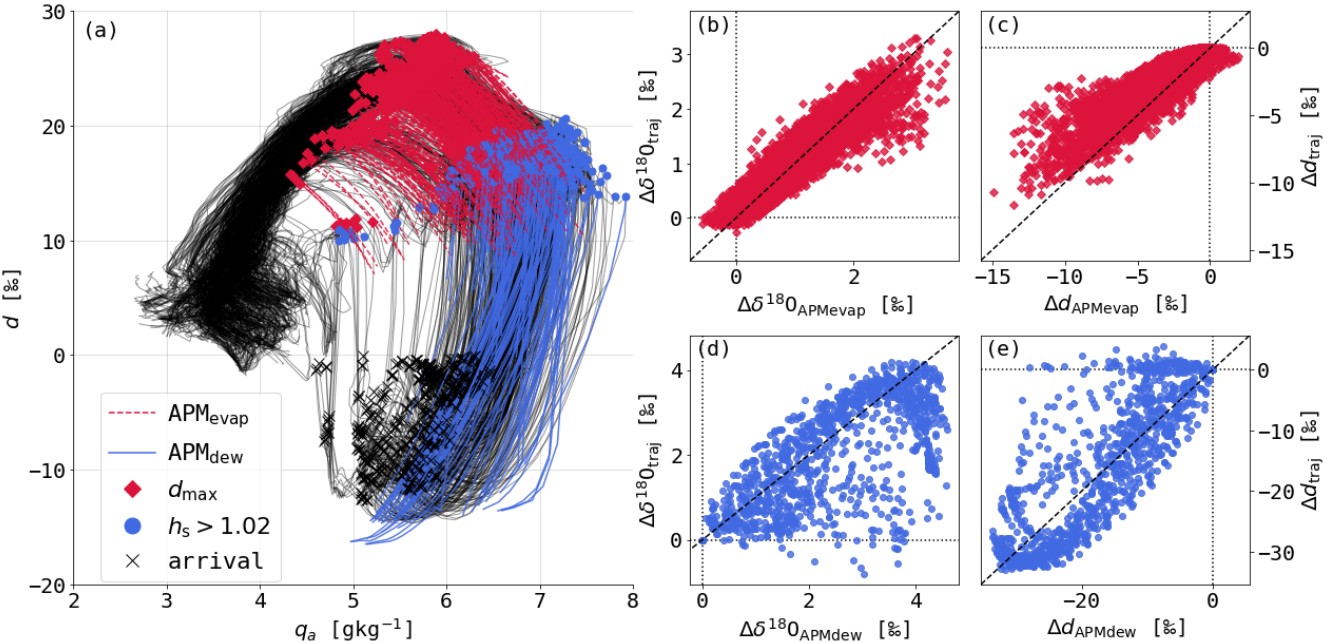

**Figure 10. (a)** Same as Fig. 9c but showing trajectories as black lines and the corresponding $APM_{dew}$ (blue solid lines) and $APM_{evap}$ (pink dashed lines) simulations using the SST and $q_a$ evolution of the $COSMO_{iso}$ trajectories. Markers denote specific events along the trajectories: the starting position of $APM_{evap}$ at highest $d$ (pink diamond), the starting position of $APM_{dew}$ at the first instance when $h > 1.02$ along the trajectories (blue dot) and the arrival of the trajectories (black cross). Comparison of changes in $\delta^{18}O$ **(b,d)** and $d$ **(c,e)** along $COSMO_{iso}$ trajectories (traj) and during the corresponding $APM_{evap}$ **(b,c)** and $APM_{dew}$ **(d,e)** simulations are shown. The changes in $\delta^{18}O$ and $d$ are shown relative to the initial composition of the APM simulations in 1-hourly resolution.

trajectories arriving in region WF are used for $APM_{evap}$ and $APM_{dew}$ simulations shown in Fig. 10a. $\Delta q_a$ in these APM simulations is defined as the change in $q_a$ during a 1-hourly time step along the $COSMO_{iso}$ trajectories. $APM_{evap}$ simulations start at the time of highest $d$ for each trajectories and run until $h_s > 0.9$. $APM_{dew}$ simulation are initiated at the first time step of $h > 1.02$ and run until $h_s < 1.005$. Additionally, if the trajectory ascends above 50 m a.s.l. the APM simulation are ended. With the SST and $q_a$ evolution from the $COSMO_{iso}$ trajectories, the $APM_{evap}$ simulation cover the entire range of simulated $d$

values in the $COSMO_{iso}$ simulations. Similarly, $APM_{dew}$ simulates lower $d$ values than in the prescribed simulations. The changes relative to the intial values for $d$ ($\Delta d$) and $\delta^{18}O$ ($\Delta\delta^{18}O$) during these APM simulations versus the changes along the corresponding trajectories are shown in Fig. 10b-e. $\Delta\delta^{18}O_{APM}$ agrees well with $\Delta\delta^{18}O_{traj}$ for both APM simulations with a larger spread for $APM_{dew}$. While $\Delta d$ agrees well for $APM_{dew}$, $\Delta d$ is a few ‰ lower in the $APM_{evap}$ simulations compared to the trajectories.

The application of the single-process APMs shows that the decrease in $d$ along the near-surface $COSMO_{iso}$ trajectories arriving in the warm sector is dominated by two processes: (1) ocean evaporation during a movement over a negative SST gradient, and (2) dew deposition in the warm sector of a midlatitude cyclone. Thus, the water vapour fluxes at the ocean-atmosphere





interface can explain most of the $d$ signal in water vapour along the trajectories. The remaining differences between the trajectories and APM simulation can have several causes. With the single-process APMs, only one process is taken into account

at a time. In reality and in COSMO$_{iso}$, the air parcel can be influenced by more than a single-process at a time, leading to complex non-linear interactions between the different processes at play. For example, turbulent and vertical transport during ocean evaporation can mix water vapour with a different isotopic composition into the air parcel. This can be expected during strong ocean evaporation in a vertically unstable atmosphere and plays a role in an evaporative environment, as for example in the subtropics (Benetti et al., 2014). In APM$_{evap}$ such conditions are mainly given in the beginning of the simulation when $h_s$

is low and strong ocean evaporation occurs in a conditionally unstably stratified MBL. The overestimation of the decrease in $d$ in the APM$_{evap}$ simulations could thus be caused by neglecting the mixing of the air parcel's water vapour with surrounding water vapour. In the oversaturated environment during dew deposition, the atmosphere is generally stably stratified. Still, due to the strong vertical wind shear, turbulence can be produced even in this stably stratified environment and might have caused vertical mixing in the warm sector's MBL. While upward turbulent transport could explain the vertical extent of the negative

$d$ anomaly, strong vertical mixing rather counteracts the formation of low $d$ near the ocean surface by mixing high $d$ from the free troposphere downward. Thus, other processes have to be considered to explain the difference between the APM$_{dew}$ simulations and the COSMO$_{iso}$ trajectories. Below-cloud processes such as the evaporation of rain droplets or the equilibration of rain droplets with the surrounding water vapour might affect SWIs in the warm sector. If the rain droplets have experience substantial evaporation beforehand, $d$ of the rain droplets is expected to be low due to non-equilibrium fractionation during

evaporation. This can introduce low $d$ into the MBL during precipitation. The near-surface trajectories of this study do not show different isotopic signals under the presence of cloud or rain water than without the presence of hydrometeors (not shown). The observed, relatively small differences between the APM$_{dew}$ simulations and the COSMO$_{iso}$ trajectories might therefore be a result of non-linear effects of the combination of several processes such as altering periods of ocean evaporation and dew deposition, variations in the ocean water's isotope composition, below-cloud and mixing processes.

## 5   Discussion and conclusions

The aim of this study was to better understand the processes leading to characteristic water vapour SWI signals observed in the warm sector of extratropical cyclones using air-parcel process models. As shown in a previous study based on a statistical analysis, dew deposition on the ocean surface, Rayleigh fractionation during cloud formation and weakening ocean evaporation mainly shape the isotopic composition of water vapour in the warm sector (Thurnherr et al., 2021). Here, we further

investigate these processes in a mechanistic way by simulating the specific SWI evolution due to different moist processes using single-process APMs of ocean evaporation and dew deposition on the ocean surface, when moving across a meridional SST gradient and Rayleigh fractionation during a moist adiabatic ascent. These simple process models adequately simulate the SWI evolution along COSMO$_{iso}$ backward trajectories and confirm the main processes identified in the case study of warm temperature advection in a Southern Ocean cyclone. Furthermore, these APMs give an estimate of the induced changes in $d$,

$\delta^{18}O$  $\delta^2H$ and $q$ by the represented processes.





The strongest changes in water vapour $d$ are caused by the modulation of air-sea fluxes such as decreasing ocean evaporation during the movement across an SST gradient towards lower SSTs and dew deposition, which can both lead to a decrease in $d$ in the order of 10‰. Air-sea interactions, and especially the change in $d$, depend strongly on $h_s$. For ocean evaporation, the evolution of $h_s$ is affected by the strength of the ocean evaporation and the SST. The higher $h_s$ at maximum SST the lower

the maximum $d$ gets and, thus, the stronger the decrease in $d$ becomes. The decrease in $d$ during dew deposition is mostly influenced by the initial oversaturation, which determines the strength of the non-equilibrium fractionation. The SST gradient determines the decrease in $d$ during dew deposition by determining how quickly saturation is reached. Rayleigh fractionation during a moist adiabatic ascent can lead to a decrease in water vapour $d$ one order of magnitude smaller than the air-sea interactions and is mainly governed by the initial temperature. The APMs illustrate the unique pathways associated with the simulated

processes in the $\delta$- and $d$-$q_a$, which can therefore be used to identify these processes along trajectories.

The introduced process models simplify the reality by only showing the effect of one process for each model. Furthermore, effects from changes in the SST and ocean skin layer SWI composition due to the air-sea interactions are neglected. Despite these caveats, the process models are able to reproduce the SWI evolution along trajectories for time periods when one moist process dominates the SWI variability.

Compared to previous Eulerian process models, which simulated the SWI composition of the MBL, and mainly investigated the effect of vertical moisture transport (e.g. Merlivat and Jouzel, 1979; Benetti et al., 2018; Feng et al., 2019), the Lagrangian perspective used here is important to understand the SWI variability in the extratropics, where the large-scale horizontal advection is 1-2 orders of magnitude larger than the vertical advection, and therefore strongly affects the SWI variability of water vapour. Previous studies have used Lagrangian approaches to model the SWI variability of water vapour. Pfahl and Wernli

(2009) integrated the evaporated water vapour from ocean evaporation at the moisture sources along trajectories to compare with measurements of SWIs in water vapour at the coast. We adapted this approach to more idealised situations of ocean evaporation across an SST gradient with $\text{APM}_{\text{evap}}$ which helps to understand the main factors influencing the SWI evolution during continuous ocean evaporation.

Rayleigh fractionation models have been frequently used to simulate the effect of continuous rain out on the isotopic compo-

sition of water vapour and precipitation arriving over land (e.g. Helsen et al., 2006; Galewsky et al., 2007; Bonne et al., 2014). With the $\text{APM}_{\text{ray}}$, we highlight, that Rayleigh fractionation during a moist adiabatic ascent can potentially lead to low $d$ in MBL water vapour due to the temperature dependency of the equilibrium fractionation factors with distinctly different changes in SWIs and $q_a$ than air-sea interaction.

Dütsch et al. (2018) used trajectories calculated from $\text{COSMO}_{\text{iso}}$ simulations to identify contributions of different processes

to the SWI composition of water vapour over Europe. This identification method based on changes of $q_a$ and $h$ along the trajectory gave detailed insight into the SWI variability over Europe. In this study we looked at each process from a mechanistic perspective and identified additional important processes in the MBL such as dew deposition. This study therefore helps to better constrain processes affecting the isotopic composition of water vapour over the ocean surface before the air parcel moves over terrestrial surfaces.

The case study of a warm temperature advection event showed that negative $d$ in near-surface water vapour in the warm sector





of extratropical cyclones is primarily a result of air-sea interactions. Moreover, the decrease in $d$ occurs shortly before arrival of the trajectories after the air parcels experienced an increase in $d$ in an evaporative environment. These fast and strong changes in $d$ upon arrival in the warm sector show that the low $d$ regions are not materially conserved, but are constantly rebuilt by newly arriving trajectories, which has two major implications for future studies:

1. First, this means that measurements of SWIs in water vapour close to the ocean surface, as for example conducted on research vessels, are good tracers of local to regional air-sea interactions. This has also been shown for ocean evaporation and vertical mixing processes with measurements of SWIs in the subtropical Atlantic MBL (Benetti et al., 2015). Here, we show that in the extratropics, $d$ in MBL water vapour changes quickly during transport due to air-sea moisture fluxes in either direction, i.e. due to ocean evaporation or dew deposition. These local air-sea interactions in extratropical cyclones, which are strongly driven by the large-scale temperature advection, dominate the $d$ variability in water vapour close to the ocean surface and overwrite the advected $d$ signal.

2. Second, while $d$ in water vapour close to the surface changes quickly from a Lagrangian perspective, the Eulerian signal is more persistent and shows low $d$ up to and above the MBL top. The low $d$ signal in the upper MBL might be a result of vertical mixing due to strong vertical wind shear. In the upper MBL, large-scale advection of $d$ might be more important than close to the ocean surface. To further study these complex vertical interactions of mixing and large-scale advection in the warm sector, the methods used here need to be supplemented by modelling methods that explicitly include mixing processes such as process tagging (Fiorella et al., 2021) or isotope tendencies in isotope-enabled numerical weather prediction models. Such methods would allow to thoroughly investigate the role of mixing for the advection of low $d$ in and above the MBL. Furthermore, such modelling approaches have to be combined with aircraft or drone-based high resolution vertical measurements of the vapour SWI composition. Such observations are indispensable to better constrain the processes affecting the upper part of the marine boundary layer at cloud base and in the cloud layer.

In summary, this study confirms and emphasizes, that the isotopic composition of near-surface water vapour is not only affected by ocean evaporation at the moisture source, but also by other processes, in particular dew deposition, during transport. These processes are a results of the specific path of individual air parcels and the evolution of SST and $h_s$ during transport by the large-scale atmospheric flow. It is the combination of observational and modelling data sets that allows to conduct such in-depth process-based analysis of MBL moisture cycling under the influence of large-scale flow patterns. In the future, this type of analysis could be used to quantify the contribution of single-processes to the SWI variability in the cold and warm sector of extratropical cyclones or to assess the lifetime of negative $d$ in the MBL to quantify the effect of low $d$ in near-surface water vapour on the isotopic composition of precipitation in the Southern Ocean and on Antarctica. Furthermore, the presented process models can be applied in various environmental settings to study the role of surface-atmosphere exchange processes influencing SWI variability along the large-scale flow.



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





## Appendix A: $APM_{dew}$

### A1 Model setup

We initiate an air parcel with a specific humidity $q_{a,0} > q_{s,0}$ and a supersaturation of $h_s = \frac{q_{a,0}}{q_{s,0}}$, where $q_s$ is the saturation specific humidity at SST. We assume that this air parcel only interacts with the ocean surface and does not exchange water vapour with its atmospheric environment. We let this air parcel equilibrate with the ocean surface by iteratively decreasing $q_a$ by an amount $\Delta q$ through dew deposition. Using the traditional bulk aerodynamic formulation, the depositional fluxes for the light water vapour $^l E$ and the heavy water vapour $^h E$ can be formulated as follows:

$$^l E = \Psi \cdot (q_a - q_s) \tag{A1}$$

$$^h E = \Psi \cdot \alpha_k \cdot (^h q_a - {}^h q_s) \tag{A2}$$

where $\Psi = \rho \lambda C_e U$, with $\rho$ the air density, $\lambda$ the latent heat of vaporisation, $C_e$ a non-dimensional transfer coefficient and $U$ the wind speed at $10\,\mathrm{m}$ a.s.l., $\alpha_k < 1$ the non-equilibrium fractionation factor, $^h q_a$ the specific humidity of the heavy water molecules and $^h q_s$ the saturation specific humidity of the heavy water molecules.

Equations A1 and A2 can be combined to get an expression of the isotopic ratio of the depositional flux $R_{\mathrm{flux}}$. Including the boundary condition that $^h q_s$ is equal to the saturation specific humidity at SST: $^h q_s = R_{\mathrm{oc}} \cdot \alpha_e \cdot q_s$, where $\alpha_e < 1$ is the equilibrium fractionation factor. Replacing $^h q_a$ by $R_a \cdot q_a$, we get the following equation

$$
\begin{aligned}
R_{\mathrm{flux}} = {}^h E / {}^l E &= \frac{\alpha_k \cdot (R_a \cdot q_a - R_{\mathrm{oc}} \cdot \alpha_e \cdot q_s)}{q_a - q_s} \\
&= \frac{\alpha_k \cdot (R_a \cdot h_s - R_{\mathrm{oc}} \cdot \alpha_e)}{(h_s - 1)}
\end{aligned}
\tag{A3}
$$

Equation A3 corresponds to the equation for the isotopic ratio of the evaporation flux of the CG65 model.

The isotopic composition of the water vapor is iteratively calculated with:

$$R_{a,i+1} \cdot q_{a,i+1} = R_{a,i} \cdot q_{a,i} - \Delta q \cdot R_{\mathrm{flux}}, \tag{A4}$$

$$\text{with } q_{a,i+1} = q_{a,i} - \Delta q \tag{A5}$$

And thus the isotope ratio of the vapour in the air parcel can be calculated as:

$$R_{a,i+1} = \frac{R_{a,i} \cdot q_{a,i} - R_{\mathrm{flux}} \cdot \Delta q}{q_{a,i+1}} \tag{A6}$$





## Appendix B: APM setups

For Fig. 9 the following APM setups are used: For each APM 10 different simulations are conducted, which are summarised
in Tab. B1. All simulation are initialised at a pressure of 1000 hPa. $APM_{evap}$ is stopped if $h_s$ increases above 0.9, $APM_{dew}$ if
$h_s < 1.04$, and $APM_{ray}$ if less than 2% of the initial moisture remains in the air parcel.

**Table B1.** Specifications of APM simulations in Fig. 9.

| Initial condition | $APM_{evap}$ | $APM_{dew}$ | $APM_{ray}$ |
|---|---|---|---|
| $SST_0$ [K] | 285.15, 286.15,..., 294.15 | 277.15, 277.95, ..., 284.35 | - |
| $T_0$ [K] | - | - | 280.15, 280.65, ..., 289.15 |
| $\delta^2 H_0$ [‰] | -137, -136, ..., -128 | -98 | -114, -113, ..., -105 |
| $\delta^{18}O_0$ [‰] | -19.5 | -13.375, -13.4, ..., -13.6 | -15 |
| $d_0$ [‰] | 19, 20, ..., 28 | 9.0, 9.2, ..., 10.8 | 6, 7, ..., 15 |
| $h_{s,0}$ | 0.55 | 1.2 | - |
| $\Delta q$ [$10^{-3} \cdot (q_s - q_a)$] | 1, 1.3, ..., 3.6 | 1 | - |





*Author contributions.* This paper is part of IT's PhD Thesis. IT and FA designed the study, IT carried out the simulations, analyses, and wrote the paper with feedback from FA. Both authors discussed the results.

*Competing interests.* The authors declare that they have no conflict of interest.

685 *Acknowledgements.* ACE was a scientific expedition carried out under the auspices of the Swiss Polar Institute, supported by funding from the ACE Foundation and Ferring Pharmaceuticals. The COSMO$_{iso}$ simulations were performed at the Swiss National Supercomputing Centre (CSCS) with the small production projects sm08 and sm32. The authors gratefully acknowledge Heini Wernli (ETH Zurich) for helpful discussions and his feedback on a previous version of this manuscript as well as Stephan Pfahl (Freie Universität Berlin) for his feedback on the single-process air parcel models definition.

690

*Financial support.* This research has been supported by the ACE Foundation, Ferring Pharmaceuticals and the BNP Paribas Foundation (grant no. ACE project 11) and the Swiss Polar Foundation (grant no. GLACE project DAWATEC).