# Peer review of "Disentangling the impact of air-sea interaction and boundary layer cloud formation on stable water isotope signals in the warm sector of a Southern Ocean cyclone"

_Atmospheric Chemistry and Physics, 2022_

## Author Comment (AC1)

**Review RC1 - acp-2022-12**

This is a well-written and comprehensive article that help understanding processes leading to negative water vapor d-excess observed in surface air during the ACE campaign, within the warm sector of an extra-tropical cyclone, south of South Africa.

The authors combine regional atmospheric modelling with water isotopes (COSMOiso simulation) together with 3 single-process air parcel models to understand the drivers of observed changes in water vapor isotopic composition.

They show that regions of low d-excess in surface water vapor are created by decreasing ocean evaporation and dew deposition at the ocean surface. Low water vapor d-excess close to the ocean surface is assessed to result from local air-sea interactions and to overwrite the advected d-excess signal.

I think this article allows better quantification and understanding of processes driving d-excess signal in near-surface ocean water vapor. In addition, the article structure guides the reader toward a good understanding of the authors' conclusions. I found this article very pleasant to read, with adapted figures. Consequently, I recommend this article to be published with minors revisions detailed bellow.

**Reply:** We thank the reviewer for their positive feedback, and their comments, which helped to improve the clarity of the manuscript.

**Minor comments**

**L37 : 2RVSMOW2.2** : typo? is the final **.2** right?

**Reply:** The $^2$RVSMOW2 atomic isotope ratio is multiplied by 2 because of the two possible positions of the deuterium in the water molecule (see equivalence of atomic vs. molecular ratios in Kerstel, 2004 and Iannone et al. 2010). To avoid confusion we now use the molecular isotope ratio for the standard and write 2RVSMOW2=3.1152×10$^{-4}$ , while removing the multiplication by 2 in the definition of δ$^2$H. The text was adapted accordingly.

**L169 : « αe »** is not described in the text (even if I agree it's a standard notation)

**Reply:** We added the following description:

"$\alpha_e$<1 is the equilibrium fractionation factor, $\alpha_k$≤1 the non-equilibrium fractionation factor of vapour with respect to liquid."

**L177 : « supplement Fig.S3 »** cited first, why not S1 ?

Re-number all supplement figures.

**Reply:** We adjusted the order of the supplement figures.

**L177-178 : « The simulation is initialized with qa,0=5 g kg−1 (and, thus, hs=0.5 because qs=10.0gkg−1 at 14°C), Δq=10−3·(qs − qa), δ2Ha,0=−137 ‰ and δ18Oa,0=−19.5 ‰ »**

Why this choice ?

How is chosen the Δq factor 10−3? Does it have an influence on the results ?

**Reply:** Thank you for pointing out that this needs further clarification. For this example simulation and also for the idealized simulations, which we compared to the trajectories, the initial conditions were chosen based on the following considerations:

- We chose typical values of δ$^{18}$O, δ$^2$H, $q$ and SST that we observed along the trajectories arriving in the warm sector of the discussed Southern Ocean cyclone in our COSMO$_{iso}$ simulation. More specifically, our choice was guided by conditions observed along trajectories that were calculated with COSMO$_{iso}$ wind fields. We specifically selected air parcels that experienced strong ocean evaporation.

- The proportionality factor 10$^{-3}$ (referred to as $ei$ in the following) is needed to relate the uptake increment $\Delta q$ to the vertical gradient in $q$ following the formulation of bulk surface evaporation flux parametrisation, while neglecting the effect of wind speed. $ei$ is important as it defines how quickly the parcel is saturated with respect to SST.  The larger $ei$, the faster the parcel becomes saturated and the shorter its traveling time over the SST

gradient. The COSMO$_{iso}$ trajectories used in this study experience a decrease in SST of -6.5 [-8.1 – -4.5] °C over a distance of approximately 1000km while they are taking up moisture and $d$ is decreasing (the values in the brackets denote the 5 – 95 percentile range).

A sensitivity experiment (Fig. RC1.1) shows that changes in $ei$ over several orders of magnitude leads to changes in the decrease of $d$ of less than an order of magnitude. A change in $ei$ will lead to only small changes in isotopic evolution of the air parcel. The thin red vertical lines show the value of $ei$ =10$^{-3}$ in the example simulation, which represents SST changes in the lower range of what can be observed along the COSMO$_{iso}$ trajectories (Fig. RC1.1b). Thus, the choice of $ei$ can be considered conservative in terms of the air parcel's travelling distance and perceived SST gradient. Based on this sensitivity analysis, we decided to decrease $ei$ to 5·10$^{-4}$ to increase the SST gradient for the example simulation.

[Figure]

*Fig. RC1.1: Sensitivity experiment showing the change in d (a), SST (b) and δ²H (c) for different values of ei with the same initial conditions as shown in the example simulation of APMevap, except for a lower isotopic composition of the ocean of -1.6‰ and -0.2‰ for δ²H and δ¹⁸O, respectively (following a comment by reviewer RC2). (d) shows the mean Δq during the simulations. The thin red vertical line shows the chosen ei value of 10⁻³ for the example simulation. The red shaded area denotes ei=5·10⁻⁴to 3.6 10⁻³.*

- This assumption for $\Delta q$ is only needed for the idealised APMevap simulations. For the APMevap simulations for conditions along the trajectories, $\Delta q$ is defined by the change in $q$ between two time steps diagnosed along the COSMO$_{iso}$ trajectories.

To clarify this point we added the following note:

"These initial values are chosen based on conditions observed during periods with ocean evaporation along the trajectories arriving in the warm sector of the discussed Southern Ocean cyclone in our COSMO$_{iso}$ simulation. We relate the uptake increment $\Delta q$ to the vertical gradient in $q$ following the formulation of the

bulk surface evaporation flux parametrisation, while neglecting the effect of wind speed and using a constant proportionality factor $5 \cdot 10^{-4}$. This proportionality factor determines the SST gradient perceived by the air parcel and sets the time until saturation in the air parcel is reached."

**APMdew**

**L235-236 : « The simulation is initialised with hs=1.1, which means that qa,0=6.8gkg−1, Δq=8·10−4·(qs −qa), δ2Ha,0=−98 ‰, and δ18Oa,0=−13 ‰**
.
Again, why this choice? End of APMevap ? (seems yes from Figure 4, but with different hs)

Why Δq=8·10−4·(qs −qa) ?

**Reply:** The choices for the initial conditions of the APMdew simulations are based on the same considerations as for APMevap representing an air parcel in the warm sector that experiences dew deposition while being transported over an SST gradient. $ei$ was chosen to represent the movement over a typical SST gradient by air parcels experiencing dew deposition in the warm sector. For COSMO$_{iso}$ trajectories calculated within the warm sector of the studied Southern Ocean cyclone, a typical SST change in such a situation is -1.5 [-4.0- -0.2] °C over a distance of approximately 500km.

A sensitivity experiment for different $ei$ in APMdew simulations (Fig. RC1.2) shows, similar to APMevap, that $d$ and $\delta^2H$ change by less than an order of magnitude while $ei$ changes up to two orders of magnitude. The simulated changes in isotopic composition are therefore only weakly sensitive to changes in $ei$.

[Figure]

*Fig. RC1.2: Sensitivity experiment with APMdew showing the change in (a) d, (b) SST and (c) δ2H during the simulation for different values of ei with the same intital conditions as shown in example simulation of APMdew, except for a lower isotopic*

*composition of the ocean of -1.6‰ and -0.2‰ for δ²H and δ¹⁸O, respectively (following a comment by reviewer RC2). (d) shows the mean Δq during the simulations. The red area denotes ei=8·10⁻⁴ to 10⁻³.*

We added a note on the reasoning behind the chosen initial values:

"As for APMevap, we chose the initialisation according to typical conditions observed during periods with dew deposition along the trajectories arriving in the warm sector of the discussed Southern Ocean cyclone in our COSMO$_{iso}$ simulation."

**Figure 3.h :** I was confused at the beginning between (h) above the purple line and hs in gray, maybe it's just me, it's clear for me now.

**Reply:** We exchanged SST and h$_s$ in panels 3.g and 3.h to avoid confusion.

**APMray**
**L269-270 : « Ta,0=8°C (which gives qa,0=6.7gkg−1), ΔSST=1°C, δ18Oa,0 = −15.0‰ δ2Ha,0 = −98‰ »**

Again, can you briefly explain why you choose these values ? (I can guess end of APMevap from Fig. 4)

**Reply:** For the initial conditions of APMray, values in between the end values of APMevap and the start values of APMdew, were chosen. This choice is based on the assumption that the air parcel is lifted from the marine boundary layer to higher levels before condensation occurs. Due to a change in the isotopic composition of the ocean (based on a comment by reviewer RC2), we slightly adjusted the starting position of APMray in the example simulation to -15‰ and -110‰ for δ¹⁸O and δ²H, respectively, and again choose values between APMevap end and APMdew intial conditions.

We added a note in the text:

"These initial values are chosen in between the end values of APMevap and the start values of APMdew,. This choice is based on the assumption that the air parcel is lifted from the marine boundary layer to higher levels before condensation occurs."

**Figure 4 :** This scheme highlights very well what you do in Section 3. Maybe you could move it at the beginning of Section 3 together with a small introduction of the APM and 3 example simulations presented after. It would help the reader to better understand the link between the 3 APMs, and also between the 3 examples (e.g. choice of start values in the examples).

**Reply:** This is a good idea. We moved Sect. 3.4 to the beginning of Section 3 thereby introducing the air parcel models with a comparison of the example

simulations as shown in Fig.4 (new Fig. 2) and followed by the more detailed discussion of the APMs. This restructuring led to only small adjustments in the text (see revised manuscript).

**Figure 5 :** Use a continuous colormap for potential temperature, unless you can justify the threshold at 294 K to separate warm and cold sectors?

**Reply:** We adjusted the colormap of Figure 5.

Is **Θe** the same as **θe** in the text ?

**Reply:** Yes, thank you for pointing out this typo, we made this consistent throughout the text.

**« The white contours show that warm temperature advection mask. »** Add information of the definition of this mask, or refer to the text.

**Reply:** We added the information on the warm temperature advection mask. We now write at line 333:

"The region of the warm sector is defined by the near-surface air-sea temperature difference as described in Thurnherr et al. (2021). If the difference between the 2m air temperature and the sea surface temperature is above 1°C, the region is associated with the advection of warm and moist air defining the warm temperature advection mask indicated by the red contours in Fig. 5. The warm sector region encompasses a triangular region in between the cold and warm front which is dominated by warm temperature advection. When referring to the warm sector in the following, we are referring to this area of warm temperature advection."

**L304 : « sharp gradients in THE »** What is THE? TPE = **θe** ? or not?

**Reply:** Yes, THE = $\theta_e$. We made this consistent throughout the text. Furthermore, we justified our choice use $\theta_e$ at 900 hPa to identify the fronts of this Southern Ocean cyclone. We now write at L. 304:

"Both fronts are visible by clear kinks in the sea level pressure contours and sharp gradients in equivalent potential temperature ($\theta_e$)at 900 hPa indicating a transition zone between two airmasses, i.e. dry and cold airmass with low $\theta_e$ and warm and moist airmasses with high $\theta_e$ (see Schemm et al. 2017 for a discussion on objective midlatitude front identification)."

**L305 :** Define **θe** in the text

**Reply:** We added this on line 324:

"Both fronts are visible by clear kinks in the sea level pressure contours and sharp gradients in equivalent potential temperature ($\theta_e$) at 900 hPa"

**Figure S1 / Figure S2 / hs in Figure S4:** Rainbow-like colormaps are to be proscribed for continuous variables, use a continuous colormap instead.

https://www.climate-lab-book.ac.uk/2014/end-of-the-rainbow/
https://mycarta.wordpress.com/2012/10/14/the-rainbow-is-deadlong-live-the-rainbow-part-4-cie-lab-heated-body/

**Reply:** We carefully reassessed the colormaps in Fig. S1, S2 and S4 and replaced them with continuous colormaps.

**L317 : « A good agreement of measured and simulated hs and qa can be seen (Fig. 6). »** I cannot see $q_a$ in Fig. 6. Can you add air temperature in Fig. 6 too ?

**Reply:** Thank you for pointing this out. The text is referring to another version of the figure. We adjusted the figure by adding specific humidity and air temperature.

**L318-320 « The simulated precipitation compares well with the measurements except for the few hours around 00 UTC on 26 December 2016, during which enhanced precipitation is simulated, while no precipitation has been measured. »**

Why focus on the 26 December 2016 00 UTC when model-observation differences are way larger from 26/12 12h ?

Model mostly underestimate precipitation, I don't understand the focus on the very show period when it is the opposite?

I would say that the first peak is well represented but the second peak is off (lower precipitation, and too late ?)

**Reply:** Thank you for pointing out this typo. The sentence is focusing on a time step which does not show the largest model-measurement differences and also lies outside of the warm advection event time period. We agree that the first peak in precipitation is represented better in the COSMO$_{iso}$ simulation than the second peak.

In our opinion precipitation is reasonably well simulated by COSMO$_{iso}$ in terms of intensity and timing. We cannot expect from COSMOiso or any other regional model to simulate precipitation exactly at the right place at the right time and with the right intensity, even in the case of a precipitation feature that is dominated by large-scale ascent such as along a front. We applied a spectral nudging of the large-scale winds above 850 hPa, but this does not prevent the model from developing mesoscale circulations, which deviate from the real

meteorology and modulate the intensity, timing and location of high-intensity precipitation cells along the front.

We adjusted the text on lines 341-343 in the following way:

"The simulated precipitation compares well with the measurements on 26 December but is shifted and shows lower intensity on 27 December 2016, during which enhanced precipitation is measured around 6 UTC, while simulated precipitation occurs after 12 UTC with lower intensity. This shift and underestimation led to too low $h_s$ and $q_a$ in the simulation in this time period."

**L340** Is **Өe** the same as above, i.e. **θe,** i.e. equivalent potential temperature at 900 hPa ?

**Reply:** Yes, we made this consistent throughout the text.

**L354-356 « Furthermore, the back-trajectories arriving in region CF, were located in region WF 48 h before arrival also coming from a region of high d with values above 20 ‰ (Fig.7a and supplement Fig. S4). »**

For CF, Fig. 7a shows low d 48h before as in Fig.S4. In Fig. S4, high d for CF is around 72h before?

**Reply:** Yes, the CF trajectories were in a region of high dexc 72h before arrival in CF and 24h before arrival in a low d region which will form region WF at 22 UTC 26 Dec 2016. We adjusted the text as follows to make this clear:

"Furthermore, the backward trajectories arriving in region CF, were located in a region of low $d$ 48h before arrival (Fig. 7a) also coming from a region of high $d$ with values above 20‰ 72h before arrival in region CF (see supplement Fig. S1a)."

**References**

Iannone, R. Q., Romanini, D., Cattani, O., Meijer, H. A. J., and Kerstel, E. R. Th. (2010), Water isotope ratio ($\delta^2$H and $\delta^{18}$O) measurements in atmospheric moisture using an optical feedback cavity enhanced absorption laser spectrometer, *J. Geophys. Res.*, 115, D10111, doi:10.1029/2009JD012895.

Kerstel, E. R. T. (2004), Isotope ratio infrared spectrometry, in *Handbook of Stable Isotope Analytical Techniques*, edited by P. A. de Groot, chap. 34, pp. 759– 787, Elsevier, Amsterdam.

Schemm, S., Sprenger, M., Martius, O., Wernli, H., and Zimmer, M. (2017), Increase in the number of extremely strong fronts over Europe? A study based on ERA-Interim reanalysis (1979–2014), *Geophys. Res. Lett.*, 44, 553– 561, doi:10.1002/2016GL071451.

Thurnherr, I., Hartmuth, K., Jansing, L., Gehring, J., Boettcher, M., Gorodetskaya, I., Werner, M., Wernli, H., and Aemisegger, F.: The role of air–sea fluxes for the water vapour isotope signals in the cold and warm sectors of extratropical cyclones over the Southern Ocean, *Weather Clim. Dynam.*, 2, 331–357, doi: 10.5194/wcd-2-331-2021, 2021.

---

## Author Comment (AC2)

**Review RC2 - acp-2022-12**

Thurnherr and Aemisegger provide a detailed, well-written manuscript that seeks to investigate the process-level causes of low vapor d-excess observed during the 2016/17 Antarctic Circumnavigation Expedition. They apply three single-process models representing impacts on isotope ratios from (a) ocean evaporation, (b) dew formation and deposition, and (c) upwind distillation, and demonstrate that these three processes follow diagnostic pathways in d18O/d-excess space. They then also compare the results from their process models to a regional NWP model simulation including isotopes to validate these models. Taken together, they suggest a larger than previously appreciated role for dew formation over the ocean for altering the d-excess of near-surface water vapor, particularly in the warm sector of extratropical cyclones.

Their analysis is rather detailed, and the process modeling provides interesting insights into the evolution of d-excess in near-surface water vapor. This paper represents a nice contribution, and only have a handful of suggestions for revision below.

**Reply:** We thank the reviewer for their positive feedback, and their comments, which helped to improve the clarity of the manuscript.

**Line-by-line notes**

1.      L. 36 – there appears to be an extra '2' in the denominator for R here.
**Reply** (as for Reviewer 1, comment 1): The $^2R_{VSMOW2}$ atomic isotope ratio is multiplied by 2 because of the two possible positions of the deuterium in the water molecule (see equivalence of atomic vs. molecular ratios in Kerstel, 2004 and Iannone et al. 2010). To avoid confusion we now use the molecular isotope ratio for the standard and write $2R_{VSMOW2}=3.1152\cdot10^{-4}$, while removing the multiplication by 2 in the definition of $\delta^2H$. The text was adapted accordingly.

2.      L. 44-46: might be good to cite a few of the observational studies that dew formation is a non-equilibrium process (e.g., Deshpande et al., 2013; Wen et al., 2012), since condensation processes are still (often) thought of as equilibrium to first order.
**Reply:** Thank you for this suggestion. We added a few references to observational studies of dew formation and mentioned that this topic has been addressed more specifically in studies over land. We changed the text as follows:
"For example, humid air that is supersaturated with respect to the sea surface temperature (SST) can experience dew deposition on the ocean surface, which is

accompanied by non-equilibrium fractionation due to the humidity gradient towards the ocean surface (Thurnherr et al., 2021). Dew deposition and the non-equilibrium fractionation effects accompanying it has been extensively studied over land as a water input into different ecosystems (e.g., Wen et al., 2012; Li et al., 2021)."

3.  L. 61-62: d can also change purely due to equilibrium effects when the Rayleigh *f* is very low (e.g., Bony et al., 2008; Dütsch et al., 2017)

**Reply:** We adjusted the sentence to include this. Actually, as we showed in Appendix A of Thurnherr et al. (2021), *d* can also be altered at higher *f* due to the temperature dependency of equilibrium fractionation. We therefore adapted the text as follows:

"During this long-range transport, *d* can change due to non-equilibrium processes or changes in the ambient temperature that impact the ratio of the equilibrium fractionation factors of $^1H_2^{18}O$ and $^1H^2HO$ (Dütsch et al., 2019, Appendix A in Thurnherr et al. 2021)."

4.  L. 104: which laser spectrometer was used and how was it calibrated?

**Reply:** We used a Picarro cavity ring-down laser spectrometer. The instrument and measurements are characterised in detail in previous studies (Aemisegger et al. 2012, Thurnherr et al. 2020). To keep this manuscript concise, we'd like to keep this section as short as possible. We changed the text as follows (at L. 111):

"For the isotope measurements a Picarro cavity ring-down laser spectrometer was used. The instrument and measurements are characterised in detail in previous studies (Aemisegger et al., 2012; Thurnherr et al., 2020)."

5.  L. 115: could the authors clarify what explicit treatment of deep convection means (i.e., is this model non-hydrostatic)?

**Reply:** Yes, COSMO$_{iso}$ is a non-hydrostatic model and we switched off all convection parametrisations (deep, mid-level and shallow convection). To explain more explicitly why we switched off all convection parametrisations, we adapted the text as follows at L. 113:

"The limited-area model COSMO$_{iso}$ (Pfahl et al. 2012) is an isotope-enabled version of the non-hydrostatic numerical weather and climate prediction model COSMO (Steppeler et al., 2003). The one-month, nudged COSMO$_{iso}$ simulation was performed for the time period 13 Dec 2016 to 12 Jan 2017 with a horizontal grid spacing of 0.125°, corresponding to ~14 km, 40 vertical levels and treating deep convection explicitly (shallow and deep convection parametrisations were switched off). The choice of treating convection explicitly at the resolution of the model grid is motivated by insights from recent studies (e.g. Vergara-Temprado et al. 2020), which show that convection parameterization schemes can be switched

off at coarser resolutions than previously thought (e.g. on the order of 10 km). Such a setup with explicit convection has been evaluated carefully by comparing it to COSMO$_{iso}$ simulations with parametrised convection and isotope observations from multiple platforms in previous studies (Dahinden et al. 2021, de Vries et al. 2022). The chosen grid spacing of 14 km allows for a large domain spanning an area of 50°x50° that is centred at 47°S, 18°E (Fig. 1a) and within which the regional model can develop its own isotope meteorology at the mesoscale, which is independent of the global model driving the COSMO$_{iso}$ simulation at the boundaries."

6.  L. 136-137: These seem to be fairly unusual choices for the isotope ratio of the ocean, could the authors clarify how these values were chosen? This is of particular note for this manuscript as it could be in part responsible for producing evaporation fluxes with a lower d-excess than might be expected. For example, using values for SMOW ($\delta_{18}O = 0$‰, $\delta_2H = 0$‰), the water undergoing evaporation has a d-excess of 0‰, but an ocean initial condition of ($\delta_{18}O = 1$‰, $\delta_2H = 1$‰) has a d-excess of -7‰, which would seem to bring down the d-excess of the evaporative flux by ~7‰ as well.

**Reply:** Thank you for pointing this out. This was not correctly stated in the manuscript. We changed the text as follows at L. 150:

"We use ECHAM5-wiso ocean surface isotope data (Werner et al. 2011), which is based on an observational dataset for $\delta^{18}O$ (LeGrande and Schmidt, 2006). The $\delta^2H$ of sea surface water is assumed to follow the relation of global meteoric waters (Craig and Gordon, 1965) and is thus equal to the $\delta^{18}O$ multiplied by a factor 8. This setting leads to ocean surface water isotope values in the study region of $\delta^{18}O=\sim-0.2$‰ and $\delta^2H=\sim-1.6$‰ and $d=\sim0$‰."

We have therefore adjusted the oceanic isotopic composition to -1.6‰ and -0,2‰ for $\delta^2H$ and $\delta^{18}O$, in our air parcel simulations as well. This change in the oceanic composition leads to relatively small changes in the APM simulations. The largest change can be seen in the evolution of *d*, which shows a smaller decrease with the new ocean composition as expected due to the higher *d* at the source, but the *d* decrease is still of the same order of magnitude as before. The figures and text have been updated accordingly.

7.  L. 169: there is often a lot of confusion regarding αk, often stemming from whether it is defined based on Di/D (and hence, αk < 1) or D/Di (hence αk > 1) (e.g., Benetti et al., 2014), where Di is the diffusivity of the isotopologue with a substituted atom ($_2H$ or $_{18}O$). Obviously, both can be correct depending on how the equations are cast, but it may be worth specifying

that you are referring to an αk value based on Di/D in your work, since the alternative definition is also widely used.

**Reply:** We added this information to avoid confusion. We now write on lines 199-200:

"αe <1 is the equilibrium fractionation factor, αk ≤1 the non-equilibrium fractionation factor of vapour with respect to liquid. "

8.    L. 235: I think the supplemental figures are not numbered in text in the order they appear.

**Reply:** We adjusted the order of the supplement figures.

9.    L. 251-252: I think this sentence could be a bit more clear – clearly rainout could play a role in altering SWIs, but it's not clear why you might expect to see these at the ocean-water interface if there has been substantial adiabatic lifting (presumably along isentropes, cf. (Bailey et al., 2019)?). Presumably this would be through mixing and/or subsidence, but it's not made clear here.

**Reply:** Thank you for pointing out that we can make this point clearer. We don't expect rainout to occur at the air-sea interface where the measurements took place. But, the isotopic composition of water vapour at the air-sea interface could still be affected by previously occurring cloud processes and downward transport by subsidence or turbulent mixing. For example, free-tropospheric air parcels entrained into the marine boundary layer might show an isotopic signal from cloud-related processes. We adjusted the text on lines 281-285 as follows:

"For air parcels close to the ocean-atmosphere interface, ocean evaporation and dew deposition are expected to be more important for the isotopic composition of water vapour than moist processes related to cloud formation at higher altitudes. Nonetheless, SWIs in near-surface water vapour might carry a signal from up-stream cloud formation, during which a decrease in $d_a$ occurred. This can happen due to near-surface fog formation or due to cloud formation in the boundary layer with subsequent downward transport by subsidence or turbulent mixing."

10.    L. 304 – is THE a misrendered θe? (Also, there appears to be some inconsistency in case: a capital Θ is used in Fig. 5 and L. 340 instead of the lower-case θ used elsewhere)

**Reply:** Yes, thank you for pointing out this typo, we made the notations consistent throughout the manuscript.

11.    L. 437-441 – this is an interesting point! In addition to the mixing process here, I wonder if the more turbulent coupling between the surface and the

near-surface atmosphere could have the effect of altering the 'effective' kinetic fractionation factor here as well and alter d independent of mixing, for example by changing the value of the exponent used on the ratio of diffusivities (eq. 5 in (Pfahl & Wernli, 2009), also (e.g., Gat, 1996; Mathieu & Bariac, 1996; Merlivat & Jouzel, 1979; Riley et al., 2002)

**Reply:** Thank you for highlighting this point. In an earlier publication (Thurnherr et al. 2020), we made use of two continuous measurements of the water vapour isotopic composition at two different heights during the Antarctic Circumnavigation Expedition. The difference between these two measurement time series showed a weak wind dependency that was interpreted as changes in vertical turbulent mixing and different relative importance of sea spray evaporation at the two elevations (see Fig. 10 in Thurnherr et al. 2020).

During the passage of the warm sector from 26 to 28 December 2016, high wind speed, moderate sea spray concentrations and a low wave age was measured near Marion Island. The measured vertical $\delta^{18}O$ gradient is close to 0‰ over 5.5 m during this period (Fig RC2.1). This could indicate strong vertical mixing close to the air-sea interface with weak influence from sea spray evaporation at both elevations (8 and 13.5 m a.s.l.). Such measurements could also be used to constrain the exponent of the ratio of diffusivities in the non-equilibrium fractionation factor as mentioned in your comment.

[Figure]

*Figure RC2.1: Temporal evolution of measured hourly sea spray concentration (blue line), 10m wind speed (grey line), wave age (black line) and difference in $\delta^{18}O$ in water vapour between measurements at 8 m and 13.5 m a.s.l. (green line) during ACE from 12 UTC 25 Dec 2016 to 12 UTC 29 Dec 2016. The vertical orange lines denote the beginning and end of the warm temperature advection event. The shaded orange areas correspond to the two periods WP1 and WP2 with low d during supersaturated conditions.*

**References**

Aemisegger, F., Sturm, P., Graf, P., Sodemann, H., Pfahl, S., Knohl, A., and Wernli, H.: Measuring variations of $\delta^{18}O$ and $\delta^2H$ in atmospheric water vapour using two commercial laser-based spectrometers: an instrument characterisation study, *Atmos. Meas. Tech.*, 5, 1491–1511, https://doi.org/10.5194/amt-5-1491-2012, 2012.

Bailey, A., Singh, H. K. A., and Nusbaumer, J. (2019). Evaluating a Moist Isentropic Framework for Poleward Moisture Transport: Implications for Water Isotopes Over Antarctica. *Geophys. Res. Lett.*, *46*(13), 7819–7827. https://doi.org/10.1029/2019GL082965

Benetti, M., Reverdin, G., Pierre, C., Merlivat, L., Risi, C., Steen-Larsen, H. C., and Vimeux, F. (2014). Deuterium excess in marine water vapor: Dependency on relative humidity and surface wind speed during evaporation. *J. Geophys. Res.*, *119*(2), 584–593. https://doi.org/10.1002/2013JD020535

Bony, S., Risi, C., and Vimeux, F. (2008). Influence of convective processes on the isotopic composition ($\delta18O$ and $\delta D$) of precipitation and water vapor in the tropics: 1. Radiative-convective equilibrium and Tropical Ocean–Global Atmosphere–Coupled Ocean-Atmosphere Response Experiment (TOGA-COARE) simulations. *J. Geophys. Res.*, *113*(D19). https://doi.org/10.1029/2008JD009942

Craig, H. and Gordon, L.: Deuterium and oxygen 18 variations in the ocean and the marine atmosphere, in: *Proceedings of the Stable Isotopes, in Oceanographic Studies and Paleotemperatures*, 1965.

Dahinden, F., Aemisegger, F., Wernli, H., Schneider, M., Diekmann, C. J., Ertl, B., Knippertz, P., Werner, M., and Pfahl, S.: Disentangling different moisture transport pathways over the eastern subtropical North Atlantic using multi-platform isotope observations and high-resolution numerical modelling, *Atmos. Chem. Phys.*, 21, 16 319–16 347, https://doi.org/10.5194/acp-21-16319-2021, 2021.

Deshpande, R., Maurya, A., Kumar, B., Sarkar, A., and Gupta, S. (2013). Kinetic fractionation of water isotopes during liquid condensation under super-saturated condition. *Geochim. et Cosmoch. Acta*, *100*, 60–72.

de Vries, A. J., Aemisegger, F., Pfahl, S., and Wernli, H.: Stable water isotope signals in tropical ice clouds in the West African monsoon simulated with a regional convection-permitting model, *Atmos. Chem. Phys.*, p. accepted, https://doi.org/10.5194/acp-2021-902, 2022.

Dütsch, M., Pfahl, S., and Sodemann, H.: The Impact of Nonequilibrium and Equilibrium Fractionation on Two Different Deuterium Excess Definitions. *J. Geophys. Res*. https://doi.org/10.1002/2017JD027085, 2017.

Gat, J. R.: Oxygen and hydrogen isotopes in the hydrologic cycle. *Annu. Rev. Earth and Planet. Sci.*, *24*, 225–62, 1996.

Iannone, R. Q., Romanini, D., Cattani, O., Meijer, H. A. J., and Kerstel, E. R. Th. (2010), Water isotope ratio ($\delta^2$H and $\delta^{18}$O) measurements in atmospheric moisture using an optical feedback cavity enhanced absorption laser spectrometer, *J. Geophys. Res.*, 115, D10111, doi:10.1029/2009JD012895.

Kerstel, E. R. T. (2004),  Isotope ratio infrared spectrometry, in  *Handbook of Stable Isotope Analytical Techniques*, edited by  P. A. de Groot, chap. 34, pp.  759– 787, Elsevier,  Amsterdam.

LeGrande, A. N., and Schmidt, G. A.:  Global gridded data set of the oxygen isotopic composition in seawater, *Geophys. Res. Lett.*, 33, L12604, doi:10.1029/2006GL026011, 2006.

Li, Y., Aemisegger, F., Riedl, A., Buchmann, N., and Eugster, W.: The role of dew and radiation fog inputs in the local water cycling of a temperate grassland during dry spells in central Europe, *Hydrol. Earth Syst. Sci.*, p. 2617–2648, https://doi.org/10.5194/hess-25-2617-2021, 2021.

Mathieu, R., and Bariac, T.: A numerical model for the simulation of stable isotope profiles in drying soils. *J. Geophys. Res. Atmos.*, *101*(D7), 12685–12696. https://doi.org/10.1029/96JD00223, 1996.

Merlivat, L., and Jouzel, J.: Global climatic interpretation of the deuterium-oxygen 18 relationship for precipitation. *J. Geophys. Res. Atmos.*, *84*(C8), 5029. https://doi.org/10.1029/JC084iC08p05029, 1979.

Pfahl, S., and Wernli, H.: Lagrangian simulations of stable isotopes in water vapor: An evaluation of nonequilibrium fractionation in the Craig-Gordon model. *J. Geophys. Res.*, *114*(D20). https://doi.org/10.1029/2009JD012054, 2009.

Pfahl, S., Wernli, H., and Yoshimura, K.: The isotopic composition of precipitation from a winter storm – a case study with the limited-area model COSMO$_{iso}$, *Atmos. Chem. Phys.*, 12, 1629–1648, https://doi.org/10.5194/acp-12-1629-2012, 2012.

Riley, W. J., Still, C. J., Torn, M. S., and Berry, J. A.: A mechanistic model of H218O and C18OO fluxes between ecosystems and the atmosphere: Model description and sensitivity analyses. *Global Biogeochem. Cy.*, *16*(4), 42-1-42–14. https://doi.org/10.1029/2002GB001878, 2002.

Steppeler, J., Doms, G., Schättler, U., Bitzer, H. W., Gassmann, A., Damrath, U., and Gregoric, G.: Meso-gamma scale forecasts using the non-hydrostatic model LM, Meteorol. *Atmos. Chem. Phys.* 82, 75–96, https://doi.org/10.1007/s00703-001-0592-9, 2003.

Thurnherr, I., Kozachek, A., Graf, P., Weng, Y., Bolshiyanov, D., Landwehr, S., Pfahl, S., Schmale, J., Sodemann, H., Steen-Larsen, H. C., Toffoli, A., Wernli, H., and Aemisegger, F.: Meridional and vertical variations of the water vapour isotopic

composition in the marine boundary layer over the Atlantic and Southern Ocean, *Atmos. Chem. Phys.*, 20, 5811–5835, https://doi.org/10.5194/acp-20-5811-2020, 2020.

Thurnherr, I., Hartmuth, K., Jansing, L., Gehring, J., Boettcher, M., Gorodetskaya, I., Werner, M., Wernli, H., and Aemisegger, F.: The role of air–sea fluxes for the water vapour isotope signals in the cold and warm sectors of extratropical cyclones over the Southern Ocean, *Weather Clim. Dynam.*, 2, 331–357, doi: 10.5194/wcd-2-331-2021, 2021.

Vergara-Temprado, J., Ban, N., Panosetti, D., Schlemmer, L., and Schär, C.: Climate models permit convection at much coarser resolutions than previously considered, *J. Clim.*, 33, 1915–1933, https://doi.org/10.1175/JCLI-D-19-0286.1, 2020.

Wen, X.-F., Lee, X., Sun, X.-M., Wang, J.-L., Hu, Z.-M., Li, S.-G., and Yu, G.-R.: Dew water isotopic ratios and their relationships to ecosystem water pools and fluxes in a cropland and a grassland in China. *Oecologia*,*168*(2), 549–561, 2012.

Werner, M., Langebroek, P. M., Carlsen, T., Herold, M., and Lohmann, G.: Stable water isotopes in the ECHAM5 general circulation model: Toward high-resolution isotope modeling on a global scale, *J. Geophys. Res.*, 116, https://doi.org/10.1029/2011JD015681, 2011.